# Photochromism from wavelength-selective colloidal phase segregation

Jing Zheng[1,2], Jingyuan Chen[2], Yakang Jin[3,4], Yan Wen[5], Yijiang Mu[2], Changjin Wu[2], Yufeng Wang[2], Penger Tong[5], Zhigang Li[3], Xu Hou[1,6,7] & Jinyao Tang[2,8 ✉]

Phase segregation is ubiquitously observed in immiscible mixtures, such as oil and water, in which the mixing entropy is overcome by the segregation enthalpy[1–3]. In monodispersed colloidal systems, however, the colloidal–colloidal interactions are usually non-specific and short-ranged, which leads to negligible segregation enthalpy[4]. The recently developed photoactive colloidal particles show long-range phoretic interactions, which can be readily tuned with incident light, suggesting an ideal model for studying phase behaviour and structure evolution kinetics[5,6]. In this work, we design a simple spectral selective active colloidal system, in which TiO₂ colloidal species were coded with spectral distinctive dyes to form a photochromic colloidal swarm. In this system, the particle–particle interactions can be programmed by combining incident light with various wavelengths and intensities to enable controllable colloidal gelation and segregation. Furthermore, by mixing the cyan, magenta and yellow colloids, a dynamic photochromic colloidal swarm is formulated. On illumination of coloured light, the colloidal swarm adapts the appearance of incident light due to layered phase segregation, presenting a facile approach towards coloured electronic paper and self-powered optical camouflage.

The macroscopic properties of materials are fundamentally determined by interactions between their basic composition units. For example, in a mixture of molecules with similar interactions, the mixing entropy dominates and leads to a well-mixed solution, whereas a distinct inter-molecular interaction leads to the enthalpy penalty and causes phase segregation[1–3,7]. Colloid solution is an excellent model system for study-ing phase transition and self-assembly at the atomic scale, in which the colloidal particles are regarded as gigantic artificial atoms[8,9]. Classical paths to phase segregation in colloidal mixtures have been demon-strated by changing thermodynamic variables such as temperature and/or solvent interactions[10–13].

On the other hand, the active matter offers a new approach for realiz-ing complex phase behaviours beyond thermodynamic equilibrium[4,14]. The segregation of active colloids has been proposed as motility-induced phase separation[15,16], in which the dispersed self-propulsion particles condense due to particles' mobility and repulsive interaction. Theo-retically, the active colloidal mixture can self-phase separate due to distinctive diffusivity[16], temperature[17] and activity[18,19].

Most recently, the light-powered microswimmers[20–22] have been developed with respect to a controllable nanorobot, which offers poten-tial for biomedical application and functional new materials[23,24] as the swimmer activity, alignment direction and interparticle interaction can be readily modulated with incident light. Owing to its flexibility, the light-powered active colloidal system has recently been applied

to active matter research[5,6], in which the particle interactions can be repeatedly turned on and off. On the other hand, one unique feature of light-powered active colloids is that the particles can be designed to respond to different wavelengths and polarization states of light[25,26], enabling selective excitation of one colloidal species within a colloidal mixture solution. Here, we present a simple spectra-selective active colloidal system composed of photosensitive TiO₂ colloidal particles suspended in redox shuttle solution. On photoexcitation, the redox reaction on TiO₂ particles generates chemical gradient, which tunes the effective particle–particle interaction. By mixing several otherwise identical TiO₂ colloidal species loaded with dyes of different absorp-tion spectra and adjusting the incident light spectra, the on-demand colloidal phase segregation is realized.

At high concentration, the light-induced segregation in this active colloidal mixture is vertically layered, corresponding to the incident light spectra, and produces a coloured colloidal enrichment accord-ingly. At the macroscopic scale, the photoactive colloidal mixture is photochromic as different colour-enriched layers are exposed. This new colour-shifting colloidal swarm relies on the rearrangement of existing pigment rather than the in situ generation of the new chromophore, which is similar to the action of pigment cells (chromatophores) in cephalopod's skin[27,28]. We expect this new programmable photochromic ink can be developed towards electronic ink[29], displays[30] and active optical camouflage[31].

[1]State Key Laboratory of Physical Chemistry of Solid Surfaces, College of Chemistry and Chemical Engineering, Xiamen University, Xiamen, China. [2]Department of Chemistry, The University of Hong Kong, Pokfulam, Hong Kong, China. [3]Department of Mechanical and Aerospace Engineering, The Hong Kong University of Science and Technology, Clear Water Bay, Hong Kong, China. [4]School of Physics, University of Electronic Science and Technology of China, Chengdu, China. [5]Department of Physics, The Hong Kong University of Science and Technology, Clear Water Bay, Hong Kong, China. [6]Department of Physics, Research Institute for Biomimetics and Soft Matter, Fujian Provincial Key Laboratory for Soft Functional Materials Research, Jiujiang Research Institute, College of Physical Science and Technology, Xiamen University, Xiamen, China. [7]Innovation Laboratory for Sciences and Technologies of Energy Materials of Fujian Province (IKKEM), Xiamen, China. [8]State Key Laboratory of Synthetic Chemistry, The University of Hong Kong, Hong Kong, China. ✉e-mail: jinyao@hku.hk

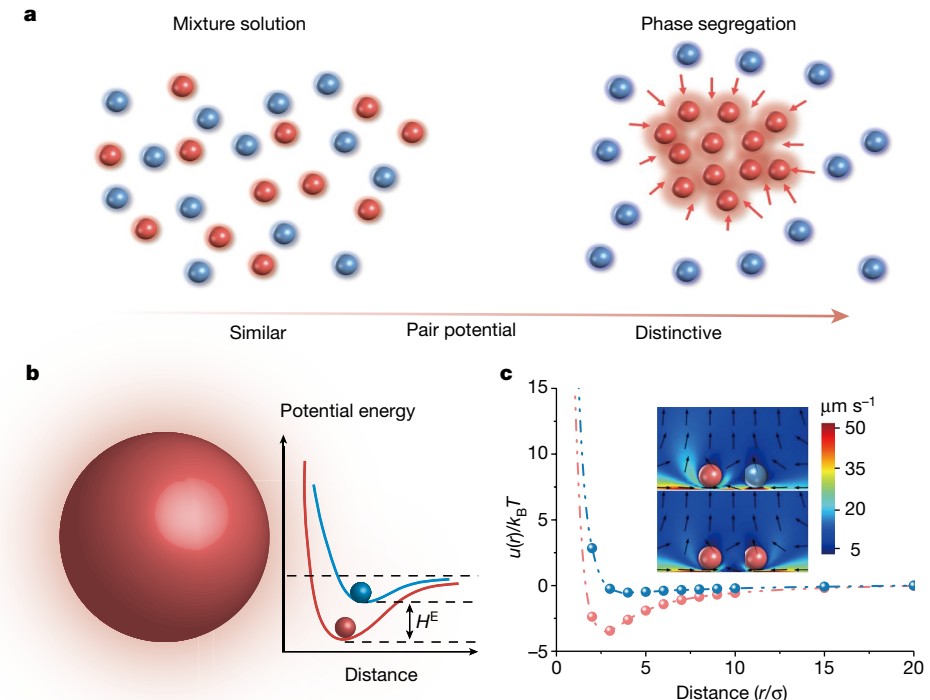

**Fig. 1 | Schematic of phase segregation in colloidal mixture. a**, Illustration of the evolution from well-mixed colloidal solution to phase segregation by adjusting the interparticle potential. **b**, Illustration of tuneable particle interaction in the dark (blue) and under photoexcitation (red), which leads to the excess enthalpy ($H^E$). **c**, Simulated phoretic flow field and effective potential of the active–active particle (red curve and bottom of the inset) and active–passive particle (blue curve and top of the inset). The particle–particle distance $r$ is normalized by particle diameter $\sigma$.

## Particle interaction tunability in spectral sensitive colloidal system

To study the phase segregation process, it is necessary to use colloids with tuneable potential. Previously, the attractive potential for colloid self-assembly[32] has been realized with polymer interactions[33], DNA strand grafting[34], depletion force[35] and so on. However, none of these approaches offers independent control of particle potential, which makes it difficult to explore the phase kinetics in the colloidal mixture. Here we adapt the simple dye-sensitized colloid to realize this interaction tuneability based on our previously demonstrated microswimmer system[25]. On photoexcitation of dyes, the redox reaction results in diffusiophoretic flow[36] and effective attractive potential. As illustrated in Fig. 1a, the segregation process can be monitored in two steps by selectively tuning the interparticle potential in the binary colloidal mixture. First, with similar particle interactions applied to two components (red and blue particles), a well-mixed dispersed phase can be prepared. Then, one component's (red particles) attractive potential is tuned to be significantly greater than another (blue particles), which contributes to the effective excess enthalpy (Fig. 1b) and leads to the segregation.

This photochemistry-powered phoretic flow has been previously used to form photoresponsive colloidal crystal[33] and active swarm[5,37,38]. Here, we denoted the unexcited particle as the 'passive' one, whereas the photoexcited particle is 'active'. The numerical simulation (Methods) is used to visualize the phoretic flow field and the interaction between active–active and active–passive particle pairs. Although the active particle interaction is non-reciprocal by nature, we assume our system's quasistatic state may be treated as a quasi-equilibrium system, which significantly simplifies the numerical treatment and facilitates understanding. As shown in the inset of Fig. 1c, the relatively long-range attraction and short-range repulsion can be predicted, which contributed to the effective attractive potential. Notably, the flow field and resulting apparent pair potential between two active particles are stronger than the active–passive pair, which contributes to the excess enthalpy for phase segregation.

In our system, $TiO_2$ colloids sensitized with spectral distinctive dyes (Methods) are used, enabling not only easy identity switching between active and passive, but also access to all the intermediate states by controlling the relative activity with illumination intensity. Experimentally, the light-induced interactions between dye-sensitized colloids as represented by the apparent pair potential $w(r)$ can be characterized by measuring the static radial distribution function $g(r)$ under varied incident light intensity. In our binary colloidal system, the dye SQ2 ($\lambda_{max}$ = 650 nm) and LEG4 ($\lambda_{max}$ = 480 nm) were selected to sensitize the $TiO_2$ colloids, respectively, then the $g(r)$ and $w(r)$ were measured in diluted conditions (area fraction $\phi \cong 1\%$) and illuminated with a wavelength variable source (Methods), whereas $w(r)$ is further fitted by standard Morse potential $u(r) = D_0[e^{-2a(r-r_0)} - 2e^{-a(r-r_0)}]$. As shown in Fig. 2a,d, $w(r)$ of the active–active particles can be tuned with incident light intensity, and the apparent bonding strength ($u_{Active-Active} = w_{min}(r)$) scales linearly with illumination intensity, whereas the bonding strength of passive–passive particles ($u_{Passive-Passive}$, Fig. 2b,e) and active–passive particles ($u_{Active-Passive}$, Fig. 2c,f) are both much less sensitive to the illumination condition. This linear dependence may be attributed to the linear photochemistry kinetics dependence on photon flux, as reported previously in a light-powered nanomotor system[39].

This unique property enables us to control the phase segregation process by tuning the particle–particle interaction with light. In classical binary mixture, the phase stability is determined by the excess enthalpy, $H^E = n\xi RT x_A x_B$, where $\xi = \frac{z}{k_B T}\left(u_{AB} - \frac{u_{AA} + u_{BB}}{2}\right)$ is a dimensionless parameter representing the interaction potential difference

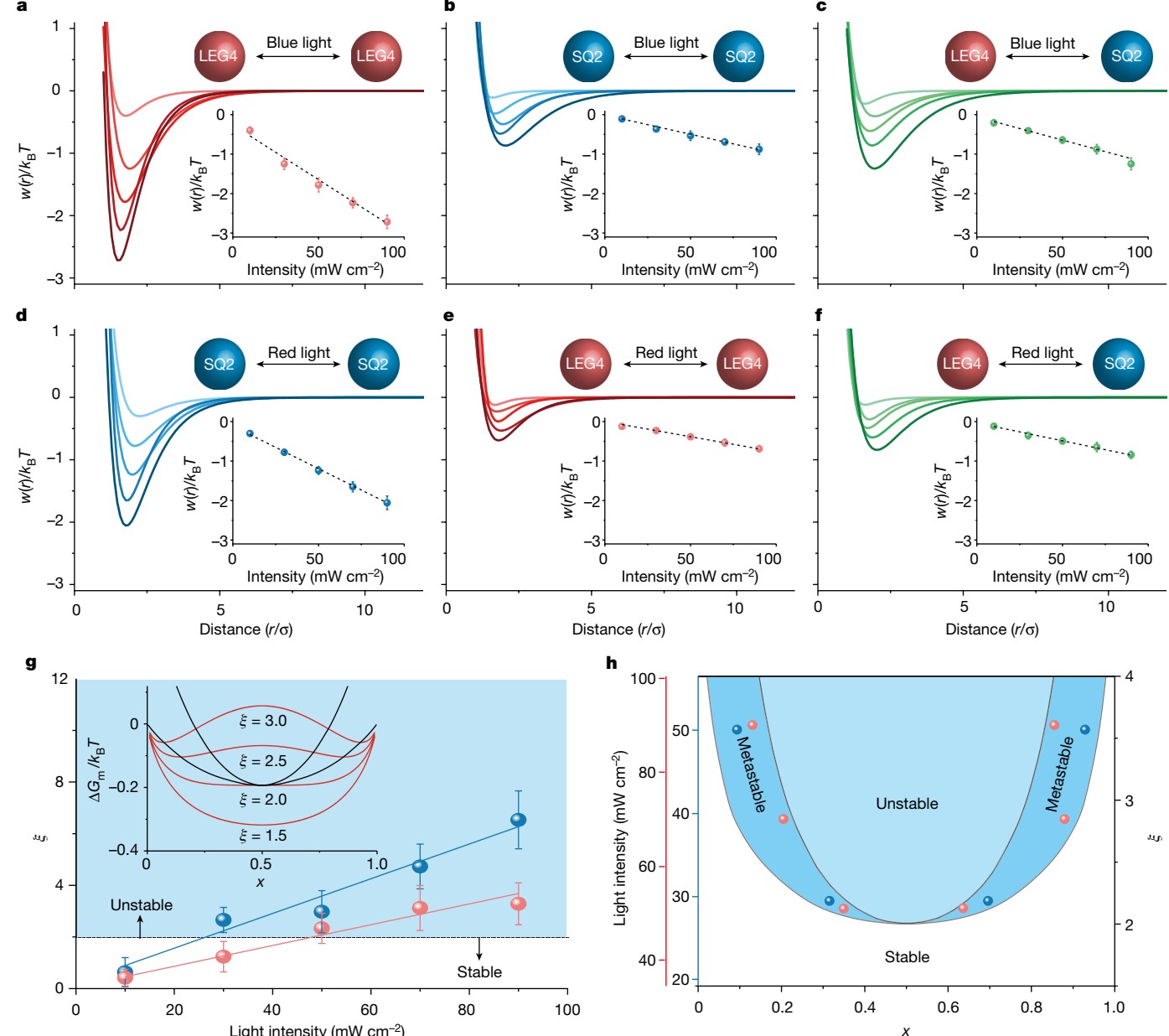

**Fig. 2 | Spectral selective effective potential of dye-sensitized-colloid modulated with illumination. a–c**, The fitted apparent pair potentials of LEG4–LEG4 sensitized (**a**), SQ2–SQ2 sensitized (**b**) and LEG4–SQ2 sensitized TiO₂ colloids (**c**) under blue light (440–480 nm) illumination with various intensity (10, 30, 50, 70 and 90 mW cm⁻²). **d–f**, The fitted apparent pair potentials of SQ2–SQ2 sensitized (**d**), LEG4–LEG4 sensitized (**e**) and LEG4–SQ2 sensitized TiO₂ colloids (**f**) under red light (640–660 nm) illumination with various intensities (10, 30, 50, 70 and 90 mW cm⁻²). The particle–particle distance $r$ is normalized by particle diameter $\sigma$. Insets show the linear dependence of

particle's bonding strength to the illumination intensity. **g**, The $\xi$ dependence on blue (blue line) and red (red line) light intensity, where $\xi = 2$ is the phase stability boundary. Inset shows the $\xi$ variation of the Gibbs energy of the binary mixture. **h**, The theoretical phase diagram of the active binary mixture with respect to red and blue light intensities (left axis) and $\xi$ (right axis). Inset points show the obtained phase compositions as observed experimentally, where red and blue points represent the condensed phases with red and blue light illumination, respectively.

between two components, $n$ is the number of particles, $R$ is the gas constant, $T$ is the temperature, $z$ is the effective coordination number, $k_\mathrm{B}$ is the Boltzmann constant, $u$ is the apparent bonding strength, and $x_\mathrm{A}, x_\mathrm{B}$ are mole fractions of A and B, respectively. In our active binary mixture, the $\xi$ increases linearly with the illuminated light intensity (Fig. 2g). As the particle's Brownian motion, corresponding to the apparent system temperature, is not enhanced proportionally (Extended Data Fig. 1), the overall effective molar free energy of segregation $\Delta_\mathrm{mix}G_\mathrm{m} = RT(x_\mathrm{A}\ln x_\mathrm{A} + x_\mathrm{B}\ln x_\mathrm{B} + \xi x_\mathrm{A}x_\mathrm{B})$ turns negative with increasing $\xi$

under strong illumination, which leads to phase segregation. When $\xi > 2$, the overall effective free energy has a double minimum (inset of Fig. 2g), leading to phase instability. As shown in Fig. 2h, the phase diagram representing the phases under various illumination intensities (scales with $\xi$) can be constructed. In this phase diagram, the outer line is the minimum $\Delta_\mathrm{mix}G_\mathrm{m}$ line corresponding to the coexistence phase, whereas the inner spinodal curve ($\frac{\mathrm{d}^2 G}{\mathrm{d}x^2} = 0$) defines the metastable mixture phase. The phase segregation is expected within the spinodal curve, as will be discussed later in our experiment.

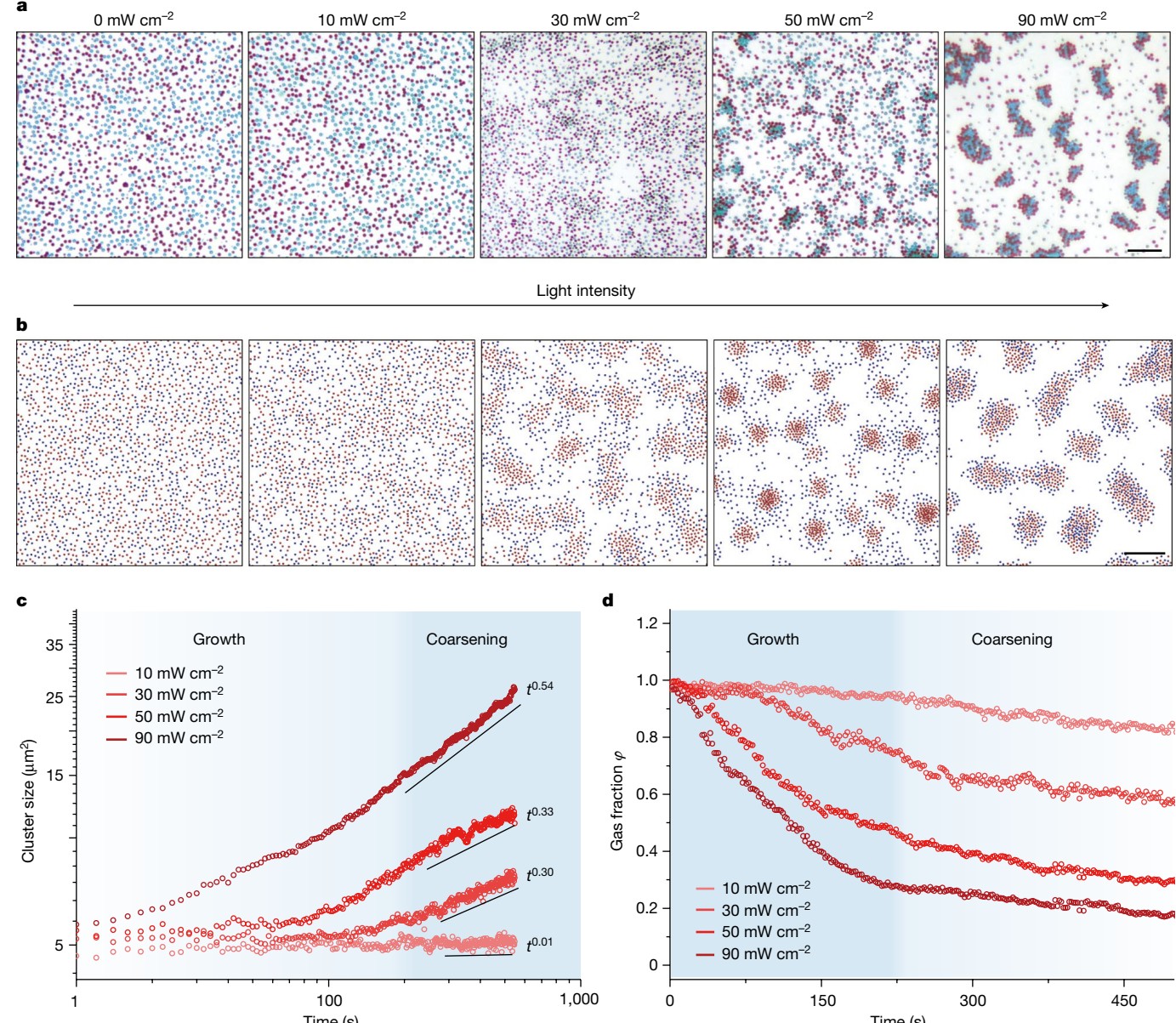

**Fig. 3 | Experimental and Brownian dynamics simulation of light-induced binary mixture phase segregation. a**, Experimental evolution of active particle (SQ2 sensitized TiO$_2$ colloid) and passive particle (LEG4 sensitized TiO$_2$ colloid) under various red light illumination levels (640–660 nm and 10, 30, 50 and 90 mW cm$^{-2}$). Scale bar, 10 μm. **b**, The Brownian dynamic simulation result of active (red) and passive (blue) particle binary mixture evolution under apparent pair potential fitted from experimental result. Scale bar, 20 μm. **c**, Experimental results of cluster size evolution of the binary colloidal system show increasing exponents with higher illumination intensity. **d**, The gas fraction of the binary mixture decreases over time under different light illumination conditions, showing the growth and coarsening stage of the phase evolution.

## Spectral selective light-induced phase segregation kinetics

To study the phase kinetics of colloidal mixture, the well-mixed colloidal solution with a one to one LEG4 to SQ2 ratio was prepared and confined in a rectangle capillary with a medium area fraction ($\phi \approx 15\%$), and subjected to uniform illumination to activate one colloidal species selectively. As shown in Fig. 3a and Supplementary Video 1, under dark conditions, both LEG4 and SQ2 sensitized TiO$_2$ colloids show indistinctive potential with apparent bonding strength less than 0.5 $k_BT$, which leads to the colloidal mixture behaving as a well-mixed gas. With increasing red light illumination (640–660 nm and 10, 30, 50 and 90 mW cm$^{-2}$), the SQ2 sensitized TiO$_2$ colloids were tuned from a passive to increasingly active state, while the LEG4 sensitized TiO$_2$ colloids remained passive due to their much weaker absorption. As stated previously, ξ scales with the light intensity (Fig. 2g), which leads to phase instability under high illumination. On the other hand, because the passive–passive (LEG4–LEG4) particle bonding strength is less sensitive to the illumination condition (Fig. 2e), and passive particle condensation is only obvious under high illumination intensity. It is worth noting that under all-illumination conditions, the observed phase composition is close to the metastable spinodal curve, which aligns with the thermodynamics of binary mixture.

To facilitate the understanding of the observed segregation dynamics, we use Brownian dynamics simulations to corroborate our experiment and test whether the non-equilibrium nature plays an essential role in the observed segregation. In our Brownian dynamics simulation, the potential parameters are adopted by fitting the

experimentally measured $w(r)$ with standard Morse potential (Supplementary Tables 1 and 2). In the simulation, all particles diffuse freely in two-dimensional (2D) space and are only subjected to the aforementioned effective potential. As no further non-reciprocal interactions, such as self-propulsion and diffusiophoresis from chemical gradient, are considered, the Brownian dynamics simulation represents the simplest physical pictures in the thermodynamic system. As shown in Fig. 3b and Supplementary Video 2, the Brownian dynamics simulation successfully catches the essence of the observed phase segregation, verified that our dye-sensitized colloid system can be regarded as the quasi-equilibrium system and may be used as a model system for the thermodynamic and kinetic study of mixture phase transition.

In active matter physics, it is prudent to ask to what degree the classic models of nucleation, growth and coarsening from irreversible thermodynamics can be applied to predict the kinetics and static properties of the active matter system[40]. For phase transition, the domain dynamics have been well-established on the basis of the premise of self-similar growth, which leads to the power-law kinetics: $S \propto t^{\nu}$ (where $S$ is the averaged size of the cluster, $t$ is the time and $\nu$ is growth exponent)[41]. The cluster coarsening in our binary system also follows self-similar growth mechanism (Extended Data Fig. 2). Here, we define cluster with a simple minimum separation criterion: all particles with centre-to-centre distance smaller than 1.5 particles in diameter are considered within the same cluster, and analyse our experimental data from Fig. 3a. As shown in Fig. 3c, the light-induced phase segregation indeed follows the power law after around 200 s of incubation delay. Furthermore, the gas fraction (the percentage of isolated particles in all particles) decreases immediately on illumination and gradually saturates after roughly 200 s (Fig. 3d), which can be regarded as the growth and coarsening stage of the phase evolution, respectively. We extracted the coarsening stage exponents $\nu$ of colloid segregation under different illumination conditions, as shown in Fig. 3c. Under mid-range light intensity (30 and 50 mW cm$^{-2}$), the growth exponents are stable around a third and increased to roughly 0.5 under strong illumination (90 mW cm$^{-2}$), in agreement with our simulation result (Extended Data Figs. 7 and 8), suggesting other growth mechanisms may play a role as the system is biased far away from equilibrium. A similar exponent increase has been reported previously in active and passive system[42,43], whereas further investigation will be needed to explain the exact cause in our light-activated mixture.

As stated previously, optical activation enables an attractive tuneable potential between different components, in which the identity of active and/or passive particles can be easily switched with different illumination conditions. As shown in Fig. 4a and Supplementary Video 3, the illumination gradually transited from pure blue (40 mW cm$^{-2}$) to combined blue to red (30:10, 20:20 and 10:20 mW cm$^{-2}$) and pure red (40 mW cm$^{-2}$), which gradually switched the active and/or passive identity of LEG4 and SQ2 sensitized particles. The corresponding light absorption induces the clustering of active particles, while the passive particles largely remain in the dispersed phase with the exception of the combined illumination, where both colloidal particles are active. The light-induced process under various illumination spectra can be quantified with the intensity of segregation. Here, considering a binary system composed of A and B particles, for simplicity, we define the intensity of segregation $I$ as $I = \frac{N_{AA} + N_{BB}}{N_{AA} + N_{BB} + N_{AB}}$, where $N_{AA}$, $N_{BB}$ and $N_{AB}$ represent the number of pairs for every particle's nearest three neighbouring particles. As shown in Fig. 4b, in a binary mixture of active and passive particles (that is, the system is only illuminated with red or blue light), the system quickly segregates while blending red and blue illumination at different ratios leads to slower and weaker segregation as both colloidal components are activated, which decreases the $\xi$.

As our dye-sensitized active colloids share the same energy cascade mechanism with well-studied dye-sensitized solar cell[44], a plethora of organic and metalorganic dyes are readily available with tuneable absorption. We expand our binary colloidal mixture to a ternary

system to cover the entire visible spectrum. In this case, the dye L0 ($\lambda_{max}$ = 420 nm) is used to sensitize TiO$_2$ colloid and offers an extra yellow channel. Under the optical microscope (Fig. 4c), the three sensitized TiO$_2$ particles show distinctive vivid colours as cyan (SQ2), magenta (LEG4) and yellow (L0), respectively. In this ternary mixture, phase segregation is still triggered by the aggregation of active particles on excitation with the corresponding light. A red, green, blue (RGB) light source with centre wavelengths of 450, 530 and 650 nm is used to activate different particles selectively (Fig. 4c). As shown in Fig. 4d, under blue illumination, both L0 and LEG4 sensitized TiO$_2$ colloids are activated, whereas the SQ2 sensitized TiO$_2$ remains passive, which leads to the cocondensation of L0 and LEG4 sensitized particles (yellow and magenta particles) from the mixture (left Fig. 4d). In the meantime, under green or red light illumination, the active species are switched to LEG4 or SQ2 sensitized TiO$_2$, where the corresponding condensation of magenta particles (LEG4 sensitized TiO$_2$) or cyan particles (SQ2 sensitized TiO$_2$) can be observed.

## Three-dimensional phase segregation and photochromic colloidal swarm

A cephalopod's skin demonstrates unmatched camouflage ability, in which the chromatophore can sense the environment's light condition and change its appearance with the actions of pigment cells accordingly[45]. Although intricate in nature, the colour-shifting ability of chromatophores is fundamentally based on a mechanical mechanism, where the pigment granules are folding and unfolding under the control of the radial muscles. Compared to the artificial chemistry-based photochromic material, this mechanical mechanism is more reliable and programmable as no chemical change is needed for the appearance shifting.

As shown previously, the ternary dye-sensitized-colloid mixture can respond to light spectra to induce selective segregation. Here, we further explore the colour-shifting ability of this active mixture. In a high-concentration mixture, the previously negligible vertical flow of individual active particles overlapped with each other and started to overcome the gravity and lead to layer segregation as illustrated in Fig. 5a and Supplementary Fig. 1a, which can be further validated by COMSOL and Brownian dynamics simulations (Extended Data Fig. 3). To verify the vertical stratification, the active colloidal mixture with close to unity area fraction ($\phi \approx 100\%$) is exposed to red, green and blue light, respectively, while the particle's distribution is visualized with the confocal microscope. As shown in Fig. 5b, under red light illumination, the SQ2 sensitized TiO$_2$ is active, which induces the aggregation at the bottom, while passive L0 and LEG4 sensitized colloids are carried to the top due to their weak interaction with the active colloids. Likewise, with green or blue light illumination, the mixture undergoes a similar stratification process, in which the active particles predominantly reside at the bottom and passive particles atop (Supplementary Table 3).

This layered stratification can be used to formulate active photochromic ink. It is worth noting that, at 100% area fraction, the mixture is still too dilute and the optical density of top layer colloids is far from realizing colloidal colourization, while the excessively high particle concentration also deteriorates the colourization ability as the interstitial space between particles diminishes, which is necessary for three-dimensional (3D) colloidal rearrangement. In addition, particle sizes need to be optimized to achieve high image quality and enhanced responsivity.

To formulate the colloidal swarm ink, we further increase the colloidal concentration to $C_0 = 4 \times 10^{11}$ per cm$^3$, which corresponds to area fraction $\phi \cong 3,000\%$. The ratio of L0, LEG4 and SQ2 sensitized colloids is tuned to around 1:2:1 to balance the reflectance (Supplementary Fig. 2a). The colloidal swarm ink is sealed into a parallel plate glass cell with a depth of roughly 100 μm. A modified commercial projector with bandpass filters is used to project the colour image onto the

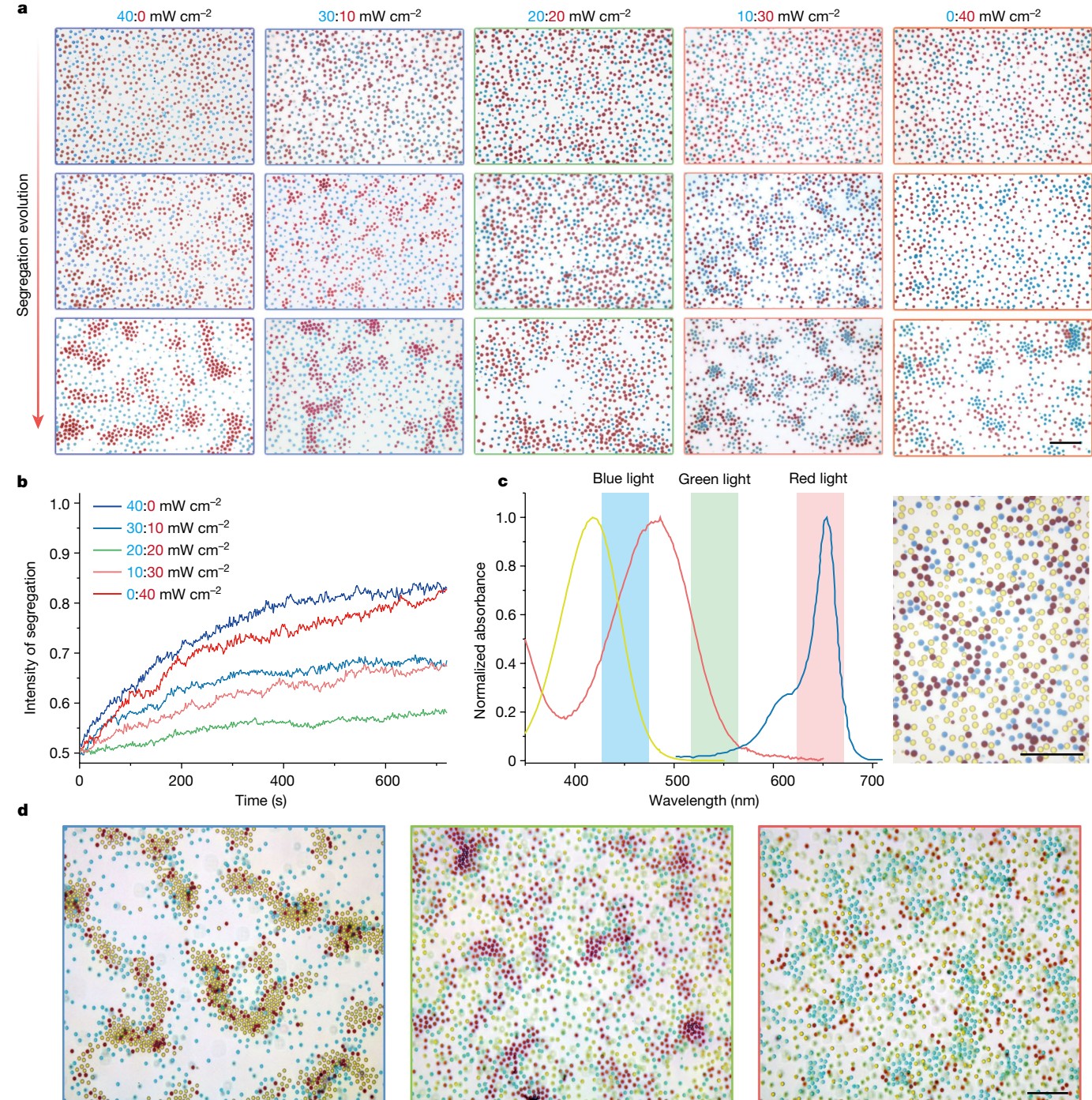

**Fig. 4 | Spectral tuneable binary and ternary mixture segregation. a**, The optical colloidal segregation evolution of LEG4 and SQ2 sensitized TiO$_2$ colloids under illumination with various blue to red ratios (40:0, 30:10, 20:20, 10:30 and 0:40 mW cm$^{-2}$). Scale bar, 10 μm. **b**, The time dependence of the intensity of segregation of the binary system under different illumination conditions. **c**, The normalized absorbance of ethanolic solution of dyes (L0 (yellow), LEG4 (magenta) and SQ2 (cyan)) with distinctive absorption spectra covering the visible spectrum. The inset shows the spectral range of illuminating RGB light source. The photograph on the right shows the colours of SQ2, LEG4 and L0 sensitized TiO$_2$ colloids under microscope. Scale bar, 10 μm. **d**, The ternary colloidal mixture composed of L0, LEG4 and SQ2 sensitized TiO$_2$ colloids shows spectra-selective phase segregation. Scale bar, 10 μm.

colloidal swarm ink to demonstrate the photochromic ability (Fig. 5c and Supplementary Fig. 2b).

First, six colour blocks of red, green, blue, cyan, magenta and yellow are projected onto the colloidal swarm ink and exposed for 120 s. As shown in Fig. 5d, the colloidal mixture acts as a positive photochromic colloidal swarm, which agrees well with our previously observed layered stratification (Fig. 5b). We further optimize the image quality

and sensitivity of the colloidal swarm ink by selecting different particle sizes and concentrations (Extended Data Figs. 4 and 5), in which the optimal performance is achieved with 500 nm particles and particle concentration at $1 \times 10^{12}$ per cm$^3$.

To demonstrate, we project a university logo on the optimized colloidal swarm with 20 mW cm$^{-2}$ illumination. As shown in Fig. 5e, an accurate colour pattern is impregnated on the colloidal swarm, and this

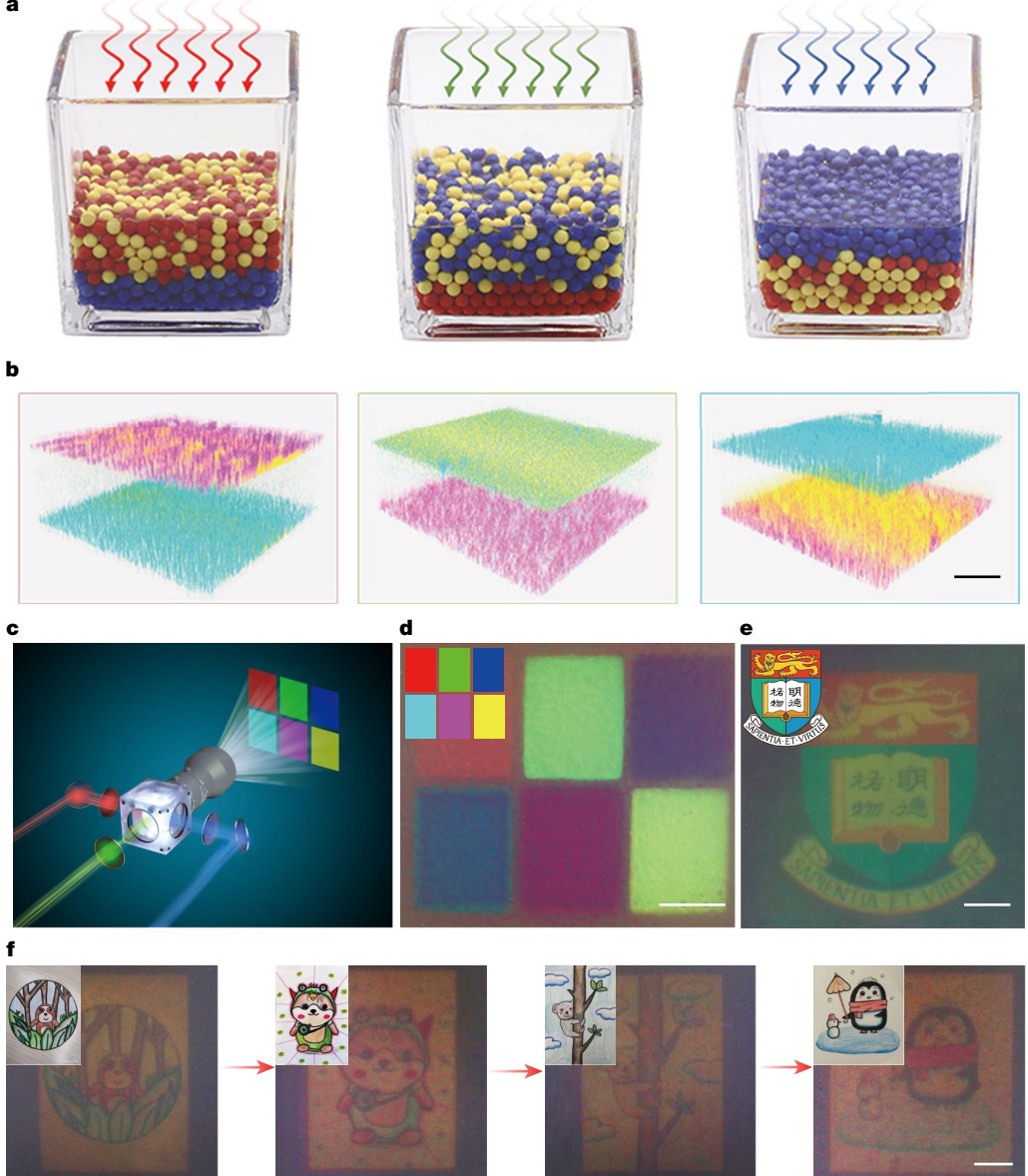

**Fig. 5 | 3D phase segregation and photochromic colloidal swarm. a**, The illustration of spectral sensitive layered segregation in the ternary colloidal system, where different illumination spectra resulted in distinctive vertical stratification. **b**, The 3D distribution of ternary colloidal particles as imaged by confocal microscope after red, green and blue light illumination. The SQ2, LEG4 and L0 sensitized $TiO_2$ colloids are represented in cyan, magenta and yellow, respectively. Scale bar, 50 μm. **c**, A modified projector is used to project designed colour images. **d**, Six colour blocks emerged on the surface of photochromic ink after 2 min of exposure. The inset shows the projected pattern. Scale bar, 2 mm. **e**, The university logo emerged on the surface of photochromic ink after 2 min of exposure. Scale bar, 2 mm. **f**, Sequential patterning of the photochromic ink with different colour paintings with 2 min exposure. The inset shows the original projected patterns. Scale bar, 2 mm.

image is quasistable up to 30 min, before the segregated colloids remix due to Brownian motion. As the colloidal swarm ink's photochromic ability is rooted in the redistribution of photoactive colloids, the colour pattern can be easily erased and repatterned with light. As shown in Fig. 5f, 120 s exposure is applied to simultaneously erase the existing image and imbed new patterns, in which various paintings are sequentially patterned into colloidal swarm ink by projecting new images over the existing ones without losing pattern quality. Furthermore, the photochromic ability of colloidal swarm ink can be further programmed by tuning the particle interactions with surface modification[46]. For instance, by controlling the surface charge polarity, the colloidal swarm ink can be programmed to show the negative colour image of the incident pattern, which manifests the flexibility of colloidal swarm ink (Extended Data Fig. 6 and Supplementary Table 4).

We summarize some key features of existing photochromic materials and compare them with our new photoactive colloidal swarm ink (Supplementary Table 5).

It is worth noting that the colloidal swarm ink's sensitivity and colour contrast could be further optimized by enhancing the colloidal swarm photoactivity with the same strategies used in dye-sensitized solar cells[47], and replacing $TiO_2$ particles with density-matched photoactive particles, in which the same design principle and physics shown in this study can be applied. On the other hand, as colloidal photochromism results from the pigment particle rearrangement, the ultimate response time should be comparable with the current e-ink system (roughly 1,000 ms), which is intrinsically slower than the LCD and OLED displays. Thus this new photochromic material is not competing with LCD or OLED technology and should target other emerging applications.

## Conclusion

In summary, we demonstrated a spectra-selective active colloidal mixture system, in which particle–particle interaction can be readily tuned by controlling the illumination spectra and intensity. The specific colloidal component can be selectively activated on excitation with corresponding light. This active mixture undergoes phase segregation that can be well predicted with phase stability thermodynamics. On the basis of this mechanism, we further demonstrate a new photochromic colloidal swarm, which self-adapts its appearance to incident light by inducing layered phase segregation. As this colloidal swarm is entirely self-sustained and simple to make, it promises wide applications from optical camouflage, smart-window for building thermal management, to full-colour electronic ink for e-readers, on further enhanced sensitivity and response time.

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

## Methods

### Synthesis of dye-sensitized monodisperse TiO$_2$ microsphere

The synthesis of the monodisperse TiO$_2$ microsphere was carried out by a modified sol-gel procedure[48]. In a typical process, 0.28 g dodecylamine was dissolved in a mixture solution containing 0.12 ml of deionized water, 50 ml of acetonitrile and 100 ml of methanol. After stirring for 5 min, 1 ml of titanium (IV) isopropoxide was added to the transparent solution and further stirred for 12 h to obtain a suspension. The suspension was centrifuged, washed with ethanol and dispersed in 15 ml of ethanol. Supplementary Fig. 3 shows the scanning electron microscopy image of the as-prepared monodisperse TiO$_2$ microsphere.

Commercial dyes SQ2 (5-carboxy-2-((3-((2,3-dihydro-1,1-dimethyl-3-ethyl-1*H*-benzo(e)indol-2-ylidene)methyl)-2-hydroxy-4-oxo-2-cyclobuten-1-ylidene)methyl)-3,3-dimethyl-1-octyl-3*H*-indolium, as received from Solaronix), LEG4 (3-(6-(4-(4-(bis(2′,4′-dibutyloxybiphenyl-4-yl)amino-)phenyl)-4,4-dihexyl-cyclopenta-(2,1-b:3,4-b')dithiophene-2-yl)-2-cyanoacrylic acid, as received from Dyenamo) and L0 (4-(diphenylamino) phenylcyanoacrylic acid, as received from Dyenamo) were used to stain the as-prepared TiO$_2$ microsphere. The absorbance of these selected dyes was measured by ultraviolet-visible light spectroscopy, showing the distinctive absorbance (Supplementary Fig. 4). In a typical staining process, 5 ml of TiO$_2$ suspension solution was centrifuged to remove the solvent and then dispersed in 5 ml fo saturated ethanolic solution of dye. The colloidal concentration can be measured by counting the number of colloids after several rounds of dilution. After 2 days of loading, the dye-sensitized TiO$_2$ colloids were centrifuged and washed with ethanol to remove excess dye and redispersed in hydroquinone (100 mM) and benzoquinone (1 mM) aqueous solution with colloidal concentration of roughly $4 \times 10^{11}$ per cm$^3$.

### Simulation of electric field and fluid flow

The particle–particle interaction is numerically simulated by the COMSOL Multiphysics package. To simulate the diffusiosmotic flow around the particle and the apparent pair potential, the interactions between the activated–passive particle pair and the activated–activated particle pair are considered. Generally, when the light with a certain wavelength is illuminated from the top, the particle is activated and the oxidation products are generated from the illuminated side. As light cannot penetrate through the particle, we only consider the illuminated hemisphere to be the oxidation reaction site, in which the chemical reaction involved is QH$_2$ → BQ + 2H$^+$ and the counter reduction reaction BQ + 2H$_2$O → QH$_2$ + 2OH$^-$ proceed at the shaded side. Owing to the different diffusion coefficients of H$^+$ and OH$^-$, an uneven distribution of charged species will be produced, resulting in a local electric field. Such self-generated electric fields drive the motion of the charged species in the electrical double layer of both the particle surface and the substrate, producing a fluid flow that drags the particle.

To build the model, three different modules will be used. The diffusion of dilute species module is used to simulate the diffusion of H$^+$ and OH$^-$, the electrostatic module is used to simulate the electric field that is generated by the competing diffusion between H$^+$ and OH$^-$, and the fluid flow module is used to simulate the diffusio-osmotic flow generated by the movement of ions in electrical double layer. For the diffusion of dilute species, the distribution of H$^+$ and OH$^-$ can be affected by the diffusion, convection and electrophoresis under the electric field. Such behaviour can be fully described by the continuity equation at a steady state,

$$\nabla \mathbf{J}_i = U \nabla \mathbf{c}_i - D_i \nabla^2 \mathbf{c}_i - \frac{z_i F D_i \nabla(\mathbf{c}_i \nabla_\varphi)}{RT} \tag{1}$$

where $J_i$ is the flux of ion $i$, $U$ is the fluidic velocity, $F$ is the Faraday constant, $\varphi$ is the electrostatic potential, $R$ is the gas constant, $T$ is the temperature and $c_i$, $D_i$ and $z_i$ are the concentration, diffusion coefficient

and charge of species $i$, respectively. The electrostatic potential ($E = -\nabla\varphi$) can be calculated from the Poisson equation,

$$-\varepsilon_0 \varepsilon_r \nabla^2 \varphi = \rho_e = F(Z_+ c_+ + Z_- c_-) \tag{2}$$

where $\varepsilon_0$ is the vacuum permittivity, $\varepsilon_r$ is the relative permittivity of water, $\rho_e$ is the charge density, $Z_+$ and $c_+$ are the charge and concentration of H$^+$ ions, and $Z_-$ and $c_-$ are the charge and concentration of OH$^-$.

The fluid flow outside the electric double layer is governed by two equations, the Stokes equation at steady state equation (3) and the continuity equation for the incompressible fluid equation (4), respectively.

$$-\nabla p + u\nabla^2 U = 0 \tag{3}$$

$$\nabla \mathbf{U} = \mathbf{0} \tag{4}$$

where $u$, $U$ and $p$ are the dynamic viscosity of water, velocity and pressure, respectively. The initial value of $U$ and $p$ are zero in our simulation.

The boundary condition on the activated particle is set to be the release or consumption of ions, representing the continuity of ion flux on the reaction hemisphere. On the reaction hemisphere, H$^+$ and OH$^-$ will both be released from the surface, so the flux of both species will be set as positive.

In the fluid flow module, the boundary condition of both the particle surface and the substrate is set to be the electro-osmotic boundary. Electro-osmotic flow is then dominated by the tangential component of the electric field,

$$E_t = E - (E - n) \cdot n \tag{5}$$

where $E_t$ is the tangential component of the electric field strength $E$. The electro-osmotic velocity is then governed by

$$u_p = -\frac{\varepsilon_r \varepsilon_0 \zeta_p}{\mu} E_t \tag{6}$$

$$u_{sub} = -\frac{\varepsilon_r \varepsilon_0 \zeta_{sub}}{\mu} E_t \tag{7}$$

$\zeta_p$, $\zeta_{sub}$ here represent the zeta potential of the particle and the substrate, $u_p$ and $u_{sub}$ are the velocity of the electro-osmotic flow on the particle and the substrate, respectively.

Extended Data Fig. 7 shows the simulation results for activated–passive (above) and activated–activated (below). The concentration distribution of H$^+$ is shown as an example of the transport of dilute species simulation. Together with the distribution of OH$^-$, the generated electric field is produced as shown in Extended Data Fig. 7b.

To calculate the apparent pair potential, we modify the model of Morse potential to be the sum of an attractive potential exerted by diffusio-osmotic flow and an electrostatic repulsion potential. The apparent attraction force exerted on the particle generated by total electro-osmotic flow is first calculated. We consider the attraction force $F_a(x)$ as a function of separation distance $x$, so that the attraction potential $u_a(x)$ can be calculated by performing an integration from the particle surface $x_0$ to $\infty$,

$$u_a(x) = -\int_{x_0}^{\infty} F_a(x) \mathrm{d}x \tag{8}$$

The attraction force at differential $x$ can be calculated by performing a volumetric integration of the pressure $P$ on the electric double layer. As our simulation is in 2D, the volumetric integration will become areal integration,

$$F_a(x) = \iint P \mathrm{d}x \mathrm{d}y \tag{9}$$

The integration area is an annular area around the particle representing the electric double layer with a thickness of 30 nm, which is estimated for an ionic concentration of $7 \times 10^{-4}$ mol m$^{-3}$. The result shows the total force exerted on a particle, we only consider the $x$ component because the $y$ component does not contribute to the attraction interaction. Attraction force on the $x$ axis is calculated by $F \times \cos(\theta)$, where $\theta$ is the included angle between the $x$ and $y$ components of the velocity. The attraction force at different separation distances can be calculated by performing a parametric sweep in COMSOL. A potential can be plotted by performing integration on the curve in commercial package Origin with equation (8),

For the repulsion potential $u_r(x)$, we adopted the equation of electrostatic repulsion between two charged spherical colloids with the Derjaguin approximation,

$$u_r(x) = \frac{Q^2}{4\pi\varepsilon_0\varepsilon_r}\left(\frac{e^{-\kappa r}}{1+\kappa r}\right)^2\frac{e^{-\kappa x}}{x} \tag{10}$$

$$Q = 4\pi r^2 \varepsilon_0 \varepsilon_r \zeta \kappa \tag{11}$$

where $Q$ is the surface charge, $\kappa$ is the inversion of Debye length, $r$ is the radius of the particle and $\zeta$ is the zeta potential of the particle.

The simulated apparent pair potential can thus be calculated by

$$w(x) = u_a(x) + u_r(x) \tag{12}$$

## Measurement of the apparent pair potential

The area fraction $\phi$ occupied by the interfacial particles is defined as $\phi = \frac{N\pi r^2}{A}$. To measure the apparent pair potential, the as-prepared LEG4-loaded TiO$_2$ and SQ2-loaded TiO$_2$ were mixed with a ratio of 1:1, and diluted 3,000 times in QH2/BQ aqueous solution. Then the mixture solution was loaded into a wax-sealed capillary (Arte Glass Associates Co., Ltd, the path length is 100 μm), with area fraction $\phi \cong 1\%$. The supercontinuum laser (SC-Pro, Wuhan Yangtze Soton Laser Co., Ltd) coupled with the variable linear filter was used as the light source. The dynamic interaction of the colloids was observed with Olympus BX51M optical microscope and recorded by a digital video camera (Canon EOS M50) at 1,920 × 1,080 resolution at 30 fps. As shown in Supplementary Fig. 5, the radial distribution function $g(r)$ can be calculated from scores of statistical location information from the recorded data, according to the general expression equation (13)[49–51].

$$g(r) = \frac{2N(r)}{A\rho^2(2\pi r dr)} \tag{13}$$

where $r$ is separation distance, $N(r)$ is the number of colloidal pairs at separation distance $r$, $N$ is the total number of colloids in each frame, $A$ is the area of the frame, $\rho = N/A$ is the number density of the particles, $2\pi r dr$ is the bin area. In this binary mixture, α and β represent the two components. The equation (14) is derived to calculate the radial distribution function between species α and β, where the number of them is denoted as $N_\alpha$ and $N_\beta$, respectively.

$$g_{\alpha\beta}(r) = \frac{A}{N_\alpha N_\beta} < \sum_{i=1}^{N_\alpha} \sum_{j=1}^{N_\beta} \delta(r - (r_i - r_j)) > \tag{14}$$

With finite concentration, the radial distribution function can reflect the interaction between neighbouring colloids. Generally, the apparent pair potential (Supplementary Fig. 6) can be determined by equation (15):

$$w(r) = -k_B T \ln g(r) \tag{15}$$

## Brownian dynamics simulation

Numerical fitting to the above apparent pair potential using homemade MATLAB codes was carried out to determine the particle–particle interaction potential and showed the standard Morse potential:

$$U(r) = D_0[e^{-2a(r-r_0)} - 2e^{-a(r-r_0)}] \tag{16}$$

where $D_0$ is the depth of the potential well, $a$ controls the 'width' of the potential and $r_0$ is the equilibrium distance. The fitting parameters will be used to describe the corresponding particle–particle interactions in Brownian dynamics simulations.

The Brownian dynamics simulations were conducted using the LAMMPS package[1]. The simulation system is 2D ($6.0 \times 6.0$ μm$^2$) and consists of 2,178 active and 2,178 passive particles. Periodic boundary conditions are used in lateral directions. The cut-off distance is set to 5.0 μm. A standard velocity-Verlet integrator with a time step of 1.0 μs is adopted to integrate the equation of motion:

$$m\frac{\partial v}{\partial t} = F_c - \frac{m}{\tau}v + F_r \tag{17}$$

where $m$ is the particle mass, $v$ is the particle velocity, $\tau$ ranging from 1.0 to 1,000.0 μs is the damping factor, $F_c$ is the conservative force from the interparticle interactions and $F_r \propto \sqrt{\frac{mk_B T}{\tau dt}}$ is the force due to solvent atoms at a temperature $T$ randomly bumping into the particle. The interaction among particles is described using the Morse potential equation (16) with the obtained parameters. Active and passive particles of the same size were uniformly dispersed at the initial state. The system was first relaxed for 2.0 s under consideration of identical particles, that is, active and passive particles are considered the same. After that, the system reached an equilibrium state, in which both active and passive particles were uniformly distributed. Then the system was run for 2.0 s ($2 \times 10^6$ time steps) using various interparticle interactions in Supplementary Table 1.

## Determination of Brownian dynamics of TiO$_2$ particles under light illumination

The SQ2 sensitized TiO$_2$ colloid was selected to illustrate the influence of light intensity on the mean squared displacement (MSD). The experiment process was similar to the measurement of pair potential, with decreased area fraction $\phi$ of 0.5%. The MSD under various red light intensities (10, 30, 50, 70 and 90 mW cm$^{-2}$) was then calculated using homemade MATLAB codes. As shown in Extended Data Fig. 1, the intensity of light had negligible effect on the MSD.

## 2D phase separation and coarsening of binary mixture

To study the phase behaviour of binary mixture, first a well-mixed solution of as-prepared LEG4-loaded-TiO$_2$ and SQ2-loaded-TiO$_2$ with a 1:1 ratio was diluted 200 times in QH2/BQ aqueous solution. Then it was sealed into a rectangle capillary and rested for 2 min to make sure all suspending colloid precipitated on the bottom surface with a roughly 15% area fraction.

To investigate the influence of light intensity on phase kinetics, red light with various intensities (640–660 nm; 10, 30, 50 and 90 mW cm$^{-2}$) was illuminated on the binary mixture solution from the top, with weak white backlight. In our definition, cluster was determined by connecting all particles with centre-to-centre separation smaller than 1.5 particle diameter, and the cluster size was averaged over all clusters in the field of view. Then the gas fraction, the percentage of isolated colloids in all colloids, was calculated. The method was also applied to the Brownian dynamics simulation of phase evolution. As shown in Extended Data Fig. 8, the simulated phase segregation result matched well with the experiment (Fig. 3) except for the growth exponent at high illumination intensity, in which the simulated growth exponent was 0.85 compared to 0.54 in the experiment. This deviation may be attributed to

the inaccurate potential of applied Morse potential, which is accurate in long range, but less accurate in the short range for describing particle–particle interaction. In high illumination, the particle–particle distance is small, which manifests this potential inaccuracy, and the growth exponents discrepancy is observed.

Furthermore, uniform light with various red-to-blue ratios (40:0, 30:10, 20:20, 10:30 and 0:40 mW cm$^{-2}$) was illuminated to the binary solution to further study the effect of incident light spectrum on the phase segregation. As shown in Fig. 4a, the various illumination spectra result in various intensities of segregation, which is defined by the following equation:

$$I = \frac{N_{AA} + N_{BB}}{N_{AA} + N_{BB} + N_{AB}} \tag{18}$$

For simplicity, the two components in the binary mixture are denoted as A and B, where $N_{AA}$, $N_{BB}$ and $N_{AB}$ represent the number of pairs for every particle's nearest three neighbouring particles.

In a ternary system, L0 sensitized $TiO_2$ colloids were introduced to offer a third colour channel. The area fraction of the ternary mixture was also tuned to 15%. The ternary phase segregation was also observed under top blue, green and red light illumination with fixed intensity (50 mW cm$^{-2}$). Obvious segregation appeared under about 2 min of top light illumination.

To quantitatively characterize the self-similarity during the phase segregation process, the chord length distribution function[43,52], $P(l/\langle l \rangle)$, was measured for a different temporal snapshot. Briefly, a straight line was randomly generated across a snapshot. The chord length $l$ was then determined by the length of the line segment inside the cluster. By varying the starting point and the orientation of the straight line randomly, a series of $l$ values from different straight lines was obtained and then normalized by the mean value (characteristic length) $\langle l \rangle$, from which the chord length distribution function $P(l/\langle l \rangle)$ was yielded.

As shown in Extended Data Fig. 2. The chord length distribution functions are independent of time, indicating a self-similar growth mechanism, which is also the origin of the power-law dependence and very similar to the phase separation in a classical thermodynamic mixture.

## 3D phase segregation

To study the vertical phase segregation of the ternary colloidal system, the 3D particle distribution was mapped by optical microscope and confocal laser scanning microscope (LEICA TCS SP8), respectively. Specifically, a colloidal mixture solution of L0, LEG4 and SQ2-loaded $TiO_2$ (the ratio for them was 1:1:1) was sealed into a parallel plated glass cell separated with parafilm ($\phi \cong 100\%$). Then the cell was exposed to the blue, green and red light (Supplementary Fig. 1a) in turn, and illuminated from the top with a fixed intensity (100 mW cm$^{-2}$). The optical microscope (the magnification and numerical aperture of the objective were ×40 and 0.9, respectively) was focused at different depths ($Z = 0$ to 80 μm), and the distribution of varying colour colloids at each depth layer was tracked. As shown in Supplementary Fig. 1b and Supplementary Video 4, the active SQ2-loaded colloids aggregated at the bottom under red light illumination, while passive L0 and LEG4-loaded colloids were pushed to the top. Conversely, under green and blue light top illumination (Supplementary Video 4), the active and passive couples were switched to LEG4/(SQ2+L0) and (L0+LEG4)/SQ2, respectively (Supplementary Table 3).

This light-inducing vertical segregation can also be visualized with a confocal microscope (Fig. 5b). Experimentally, the ternary mixture solution sealed in the homemade cell was placed under the inverted confocal microscope and exposed to blue, green and red light illumination from the top, in turn, at a fixed intensity (100 mW cm$^{-2}$). After about 2 min of illumination, the 3D confocal images were captured in which the fluorescence signals of SQ2, LEG4 and L0 were set as cyan, magenta, and yellow, respectively.

## 3D simulation of the phase segregation

To simulate the 3D phase segregation, COMSOL Multiphysics was used to simulate the detailed flow field in the particle matrix, while all the particles' top parts received higher light intensity than the bottom due to scattering and shadowing of the particle matrix. As shown in Extended Data Fig. 3a, on illumination, a vertical flow was generated between particles by diffusiophoresis, which was the origin of the attractive potential. Under this attractive potential, active particles clustered together and the vertical flows overlapped with each other and intensified (Extended Data Fig. 3b).

In the 3D phase separation experiment, there were roughly 30–50 layers of particles in the colloidal solution, and the lower particles served as the pseudo-substrate for the upper layers. As shown in simulation (Extended Data Fig. 3c), when passive particles (blue) settled below a layer of active particles (red), the upwards electro-osmotic flow was generated similarly to the particle monolayer. As more layers of active particles stacked, the upwards flow increased, which brought passive particles to the top, while the active particles made a sediment at the bottom due to the counteraction of the upwards fluid flow generation (Extended Data Fig. 3d,e).

We further explain this 3D layering process in crowded conditions with 3D Brownian dynamics simulations, where the light-dependent Morse potential, $u = D_0[e^{-2\alpha(r-r_0)} - 2e^{-\alpha(r-r_0)}]$ describing the particle interaction and the light-dependent vertical force describing the upwards flow field is adopted. Here, the parameters of potential function, that is, the depth of the potential well $D_0$ and $\alpha$ controlling the 'width' of the potential, were obtained from fitting the apparent pair potentials, as measured in our experiment (Fig. 2). Besides, according to the experimental observation and former diffusiophoresis simulations, a vertical force that varied with the average vertical distance $h$ was added, and the magnitudes of imposed vertical forces applied on the two distinct particles were proportional to the particle activity. Initially, it was assumed that active and passive particles were uniformly distributed in the simulated domain. A 3D simulation with constant temperature, volume and number of particles ($N$, $V$ and $T$) was then carried out. As shown in Extended Data Fig. 3f, active and passive particles spontaneously separated under the light illumination, which is consistent with our experimental observation.

## Macroscopic photochromic ink and characterization

To formulate the photochromic colloidal swarm ink, the as-prepared L0-loaded-$TiO_2$, LEG4-loaded-$TiO_2$ and SQ2-loaded-$TiO_2$ solutions were mixed at a 1:2:1 ratio (Supplementary Fig. 2a). A commercial 3LCD projector was chosen to project designed colour textures onto the as-prepared colloidal swarm ink. To obtain pure blue, green and red light, three optical filters were placed between the dichroic combiner cube and the original RGB light source to narrow the default broader output wavelength range for non-overlapping output (Supplementary Fig. 2b,c). To project high-resolution image, the projector lens was inverted.

In the macroscopic imaging demonstration, the colloidal swarm ink was injected into a parallel plate glass cell separated with parafilm, while the designed textures were projected onto the ink and exposed for 120 s. The obtained colour images were recorded with a digital camera (Canon EOS M50). The colour gamut was measured to characterize the performance of the photochromic ink. First, a series of images were obtained by projecting six colour blocks (Fig. 5d) with different times and intensities. Then all the samples were placed under the pinhole of the integrating sphere, and the simulated sunlight was illuminated into the integrating sphere. The spectrometer and commercial software (Oceanview) were used to draw the received reflected light spectrum into the colour gamut diagram (polytetrafluoroethylene is selected as a white Lambertian diffuser).

## Particle size dependence of photochromic ink performance

To study the influence of the particle size on the photochromic performance of the active ink, we investigated the photochromic performance of colloidal swarm with 500 nm, 1 µm and 2 µm $TiO_2$ particles, respectively. As shown in Extended Data Fig. 4, smaller particles resulted in much-enhanced photosensitivity, in which the brightness and contrast of the resulting image improved, whereas the larger particles showed much deteriorated image quality. To quantify the photosensitivity, characteristic curves of colloidal swarms with 500 nm, 1 µm and 2 µm $TiO_2$ were measured and plotted as contrast versus light intensity and contrast versus exposure time. From the characteristic curves, we concluded that the minimum light intensity and exposure time were 20 mW cm$^{-2}$ and 1 min, respectively.

We speculate that the photosensitivity is determined by the balance of upwards flow generated by active particles and the gravity of passive particles. As a result, the photosensitivity is not fundamentally limited by the working mechanism. Instead, lower density particles (such as high porosity $TiO_2$ (ref. 53) or polystyrene-$TiO_2$ core-shell particles[54]) with less gravitational drag may further enhance colloidal swarm photosensitivity. In addition, new photosensitive dyes with high absorption coefficients or better dye loading strategies, such as shown in recent publication[47] may also enhance the photochromic performance of colloidal swarm.

## Performance of the photochromic ink with different particle concentrations

Owing to thermal fluctuation, the colloidal swarm behaves as a liquid in which interstitial space allows the particle rearrangement for segregation. On the other hand, this interstitial space shrinks as particle density decreases, which prevents the segregation process. As shown in Extended Data Fig. 5, the segregation is observed in diluted ink, while the image contrast is not high as there are insufficient particles in the swarm. The colloidal swarm image quality gradually improves with particle concentration and an optimal concentration is achieved at $1 \times 10^{12}$ per cm$^3$. Further increasing particle concentration leads to deteriorated contrast and brightness, suggesting incomplete segregation.

## Performance of the photochromic ink with different zeta potentials

As the colour image of the photochromic ink is a result of phase segregation, the response can be easily tuned with simple surface modification or doped with different dyes. As shown in Extended Data Fig. 6 and Supplementary Table 4, a negative colour image is achieved with positively charged particles instead of positive colour images for negatively charged particles.

We also compared the key features of current available photochromic materials, which are mainly based on the valence transition in material or photoisomerization of chromophore molecules. As shown in Supplementary Table 5, the performance of our colloidal swarm ink offers full-colour rendering ability and immediate repatterning ability with high sensitivity and response speed[55–60].

## Data availability

The data that support the findings of this study are available from the corresponding author upon reasonable request. Source data are provided with this paper.

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

**Acknowledgements** We thank T.Y. Tang for contributing personal artwork for the photochromic demonstration in Fig. 5f. This work was supported in part by the Hong Kong Research Grants Council (RGC) General Research Fund (grant nos. GRF17307221, GRF16300421, GRF17304618 and GRF16300421), the Collaborative Research Fund (grant no. C7082-21G), RGC Research Fellowship (grant no. RFS2122-7S06), Croucher Foundation Senior Research Fellowship (2022), Croucher Innovation Award (2019), the Shenzhen-Hong Kong Innovation Circle Program (grant no. SGDX2019081623341332), the National Natural Science Foundation of China (grant nos. 22275156, T2241022, 52025132, 21975209 and 22121001), the Fundamental Research Funds for the Central Universities of China (grant no. 20720220019) and the Tencent Foundation (The XPLORER PRIZE).

**Author contributions** J.T. and J.Z. conceived and designed the project. J.Z. prepared the samples and conducted most of the measurement. Y.W. and J.Z. calculated the apparent pair potential. Y.J. performed the Brownian dynamic simulation. J.C. helped with COMSOL simulation. J.T., J.Z. and X.H. wrote the manuscript. All authors discussed the results and commented on the manuscript.

**Competing interests** The authors declare no competing interests.

**Additional information**
**Correspondence and requests for materials** should be addressed to Jinyao Tang.

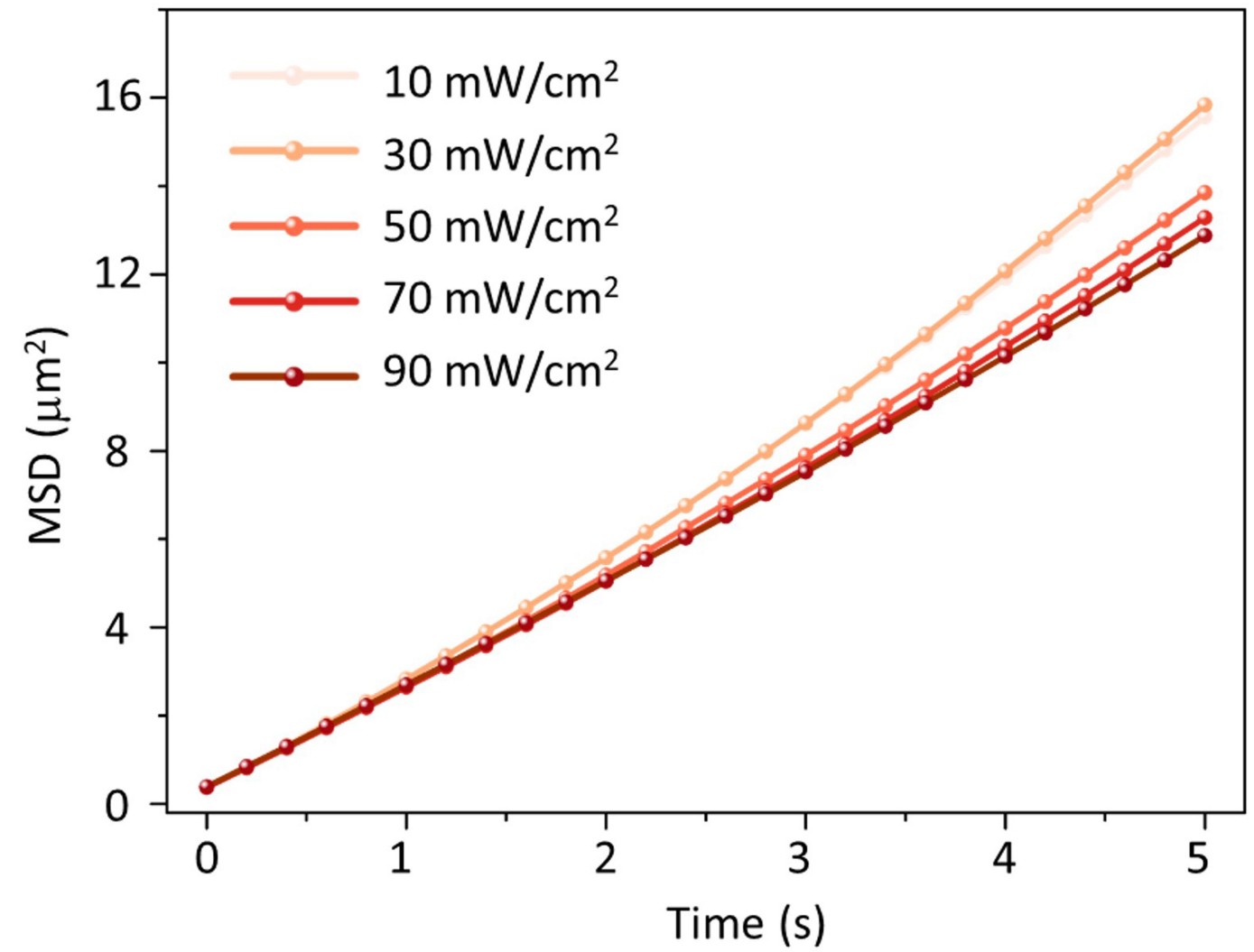

**Extended Data Fig. 1 | Mean squared displacement in varies illumination intensity.** Mean squared displacement (MSD) as a function of time for SQ2 sensitized TiO$_2$ colloids under red light (640–660 nm) illumination with various intensities (10 mW/cm$^2$, 30 mW/cm$^2$, 50 mW/cm$^2$, 70 mW/cm$^2$, 90 mW/cm$^2$).

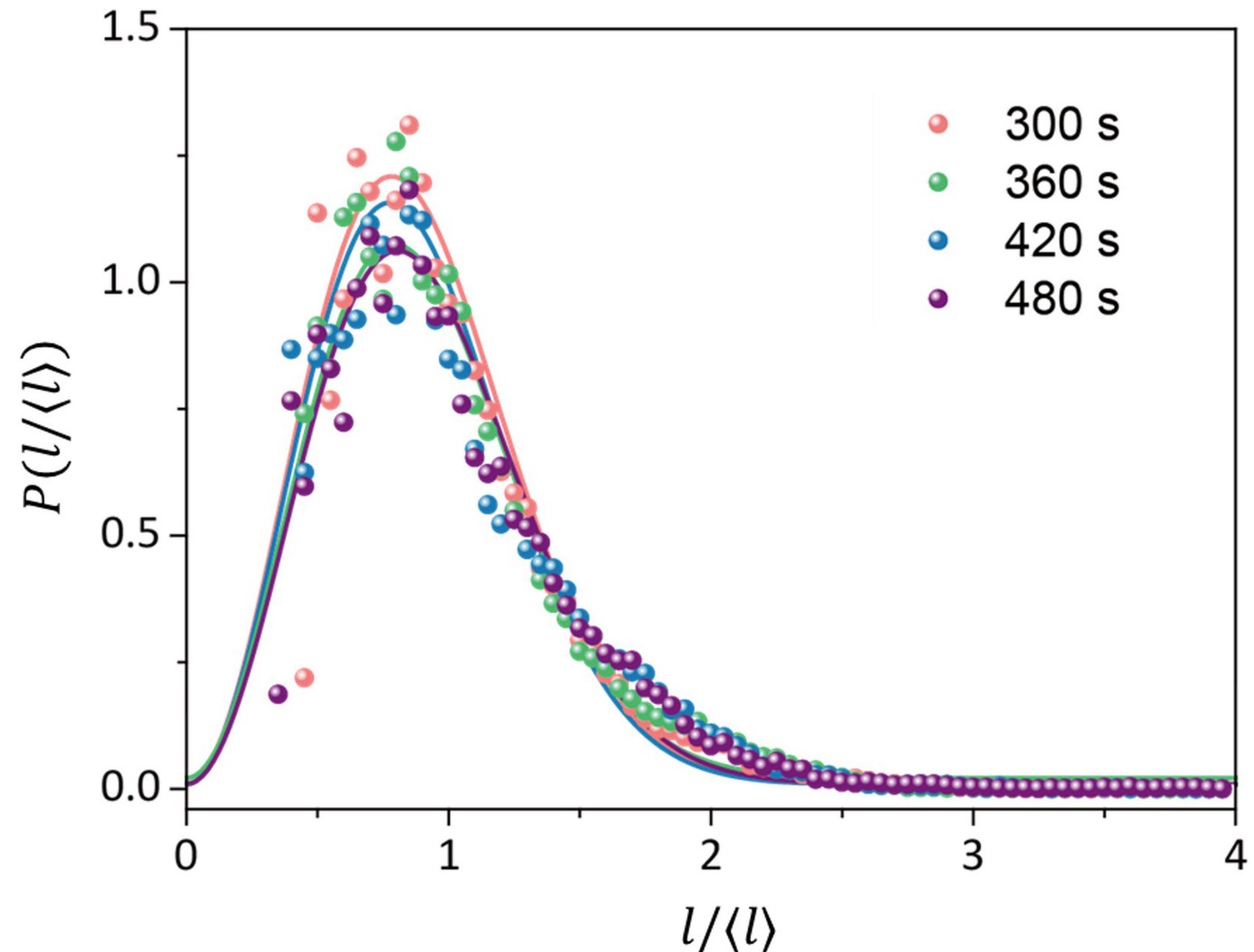

**Extended Data Fig. 2 | The time evolution of the chord length distribution.** The chord length distribution functions for the colloid-poor phase at different times (300 s, 360 s, 420 s and 480 s) after being scaled by the characteristic length ⟨l⟩.

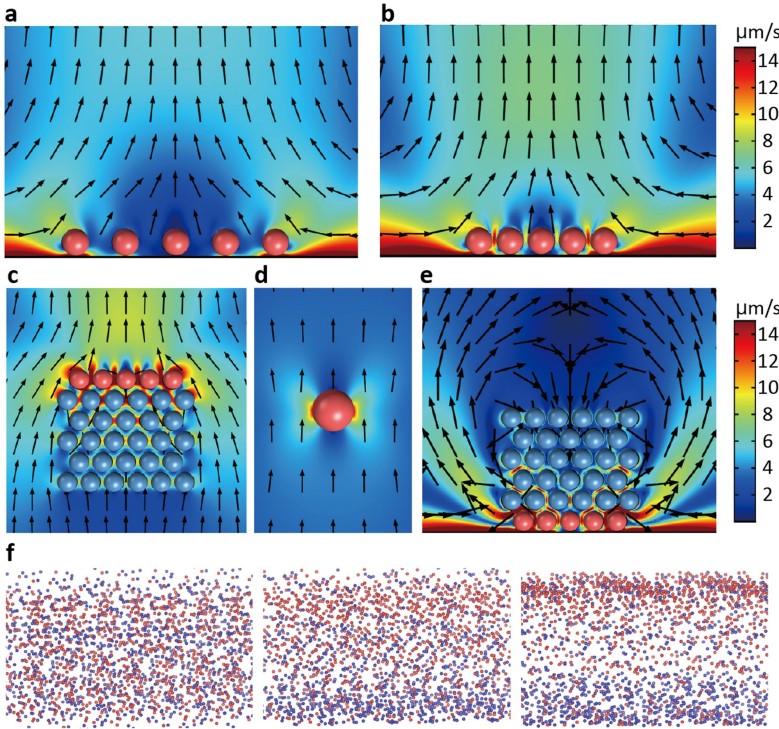

**Extended Data Fig. 3 | Simulation of Passive/Active particle mixture.**
**a**–**b**, Flow field simulation around active particles (red) and passive (blue) particles shows **a**, inward fluid flow that pushes the particles together when well separated and **b**, the upward fluid flow intensified after cluster formation. **c**–**e**, Simulations of fluid flow on particles under different conditions. **c**: Active particles located on multiple layers of passive particles, referring to the initial state of the active particles; **d**: One active particle suspended in solution, moving downwards due to the self-generated upward flow in light gradient; **e**: Active particles located below passive particles, referring to the final state of the active particles. **f.** Evolution of active (red) and passive (blue) particles in the binary mixture from 3D Brownian dynamics simulations.

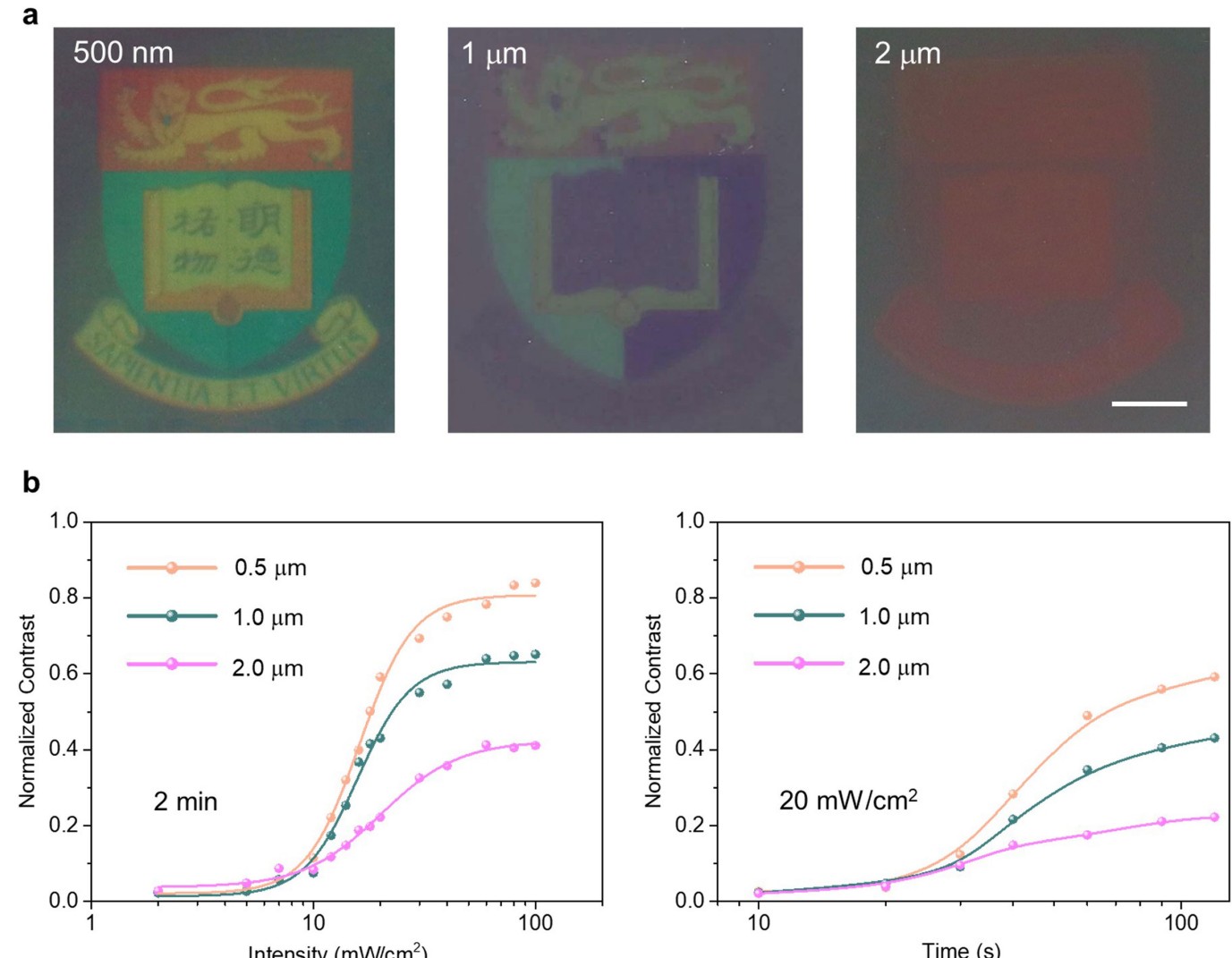

**Extended Data Fig. 4 | The performance of photochromic active colloidal with various particle diameters. a,** The optical images of the texture of different particle sizes. Scale bar: 2 mm. **b,** Characteristic curve of different photochromic nanoswarm under different light intensity and time.

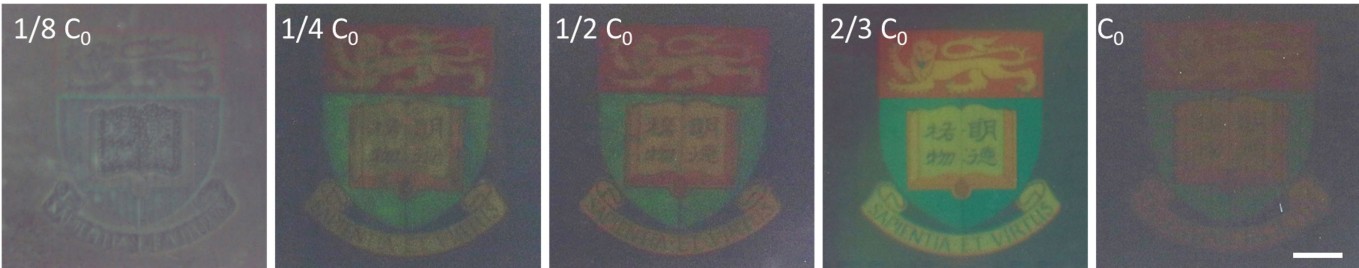

**Extended Data Fig. 5 | Concentration dependence of the photochromic property of the colloidal swarm.** performance of photonic nanoswarm with different concentrations of the particle mixture. Scale bar: 2 mm.

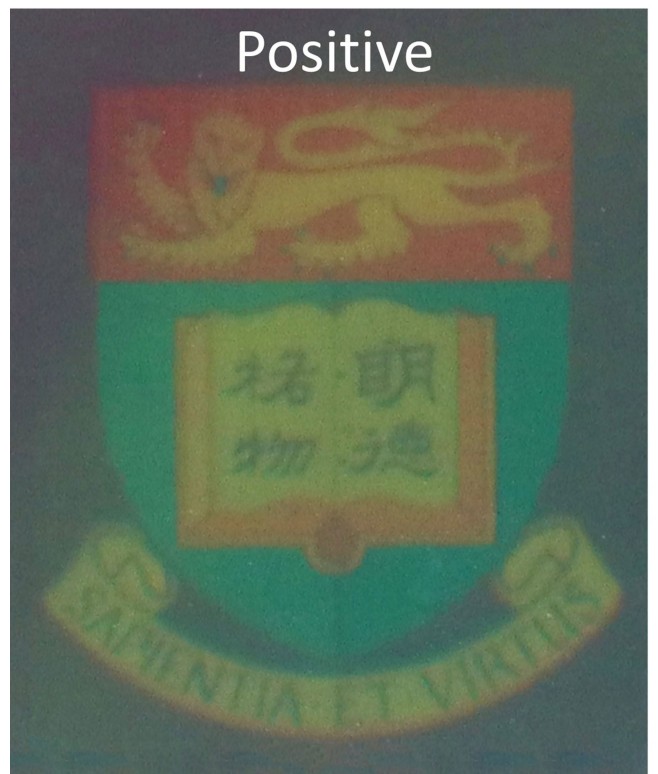
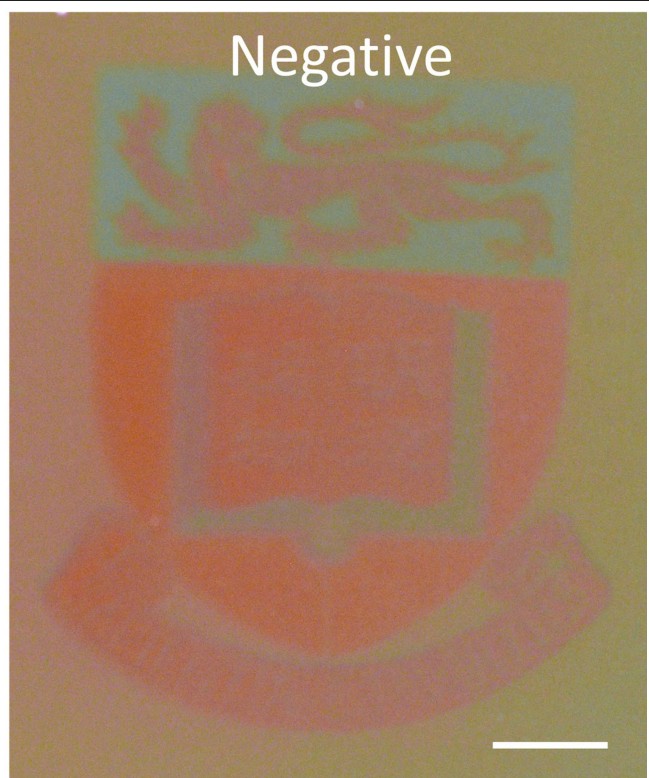

**Extended Data Fig. 6 | Photochromic colloidal swarm with different charge polarity.** Positive and negative images of photochromic nanoswarm modulated by zeta potential. Scale bar: 2 mm.

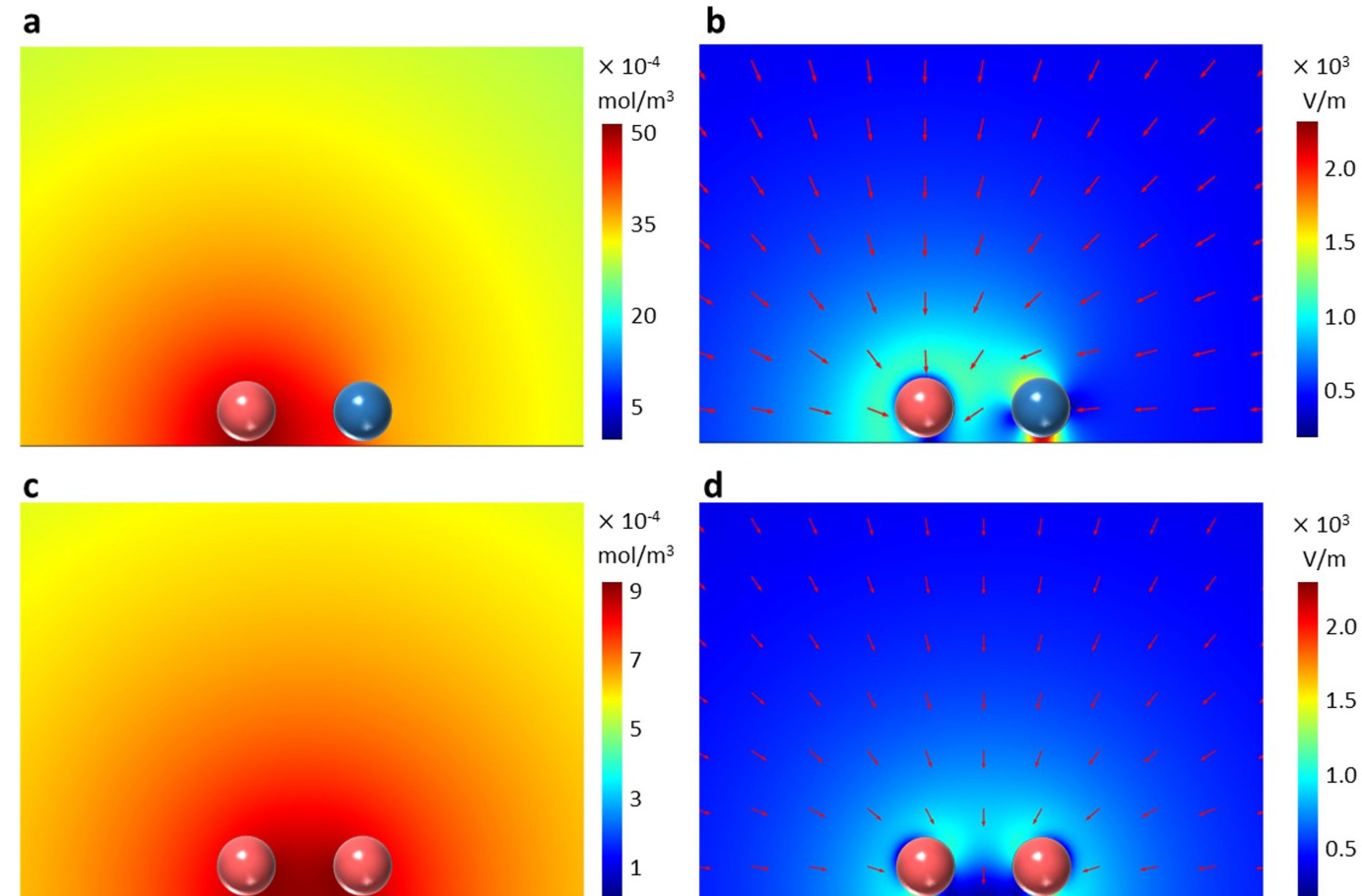

**Extended Data Fig. 7 | Ion and electric field distribution around active and passive colloidal. a**, Simulated concentration distribution of H⁺ ions produced by the chemical reaction between active and passive particles. **b**, The electric field of the active-passive species. **c**, Simulated concentration distribution of H⁺ ions produced by the chemical reaction between two active particles. **d**, The electric field of the active–active species.

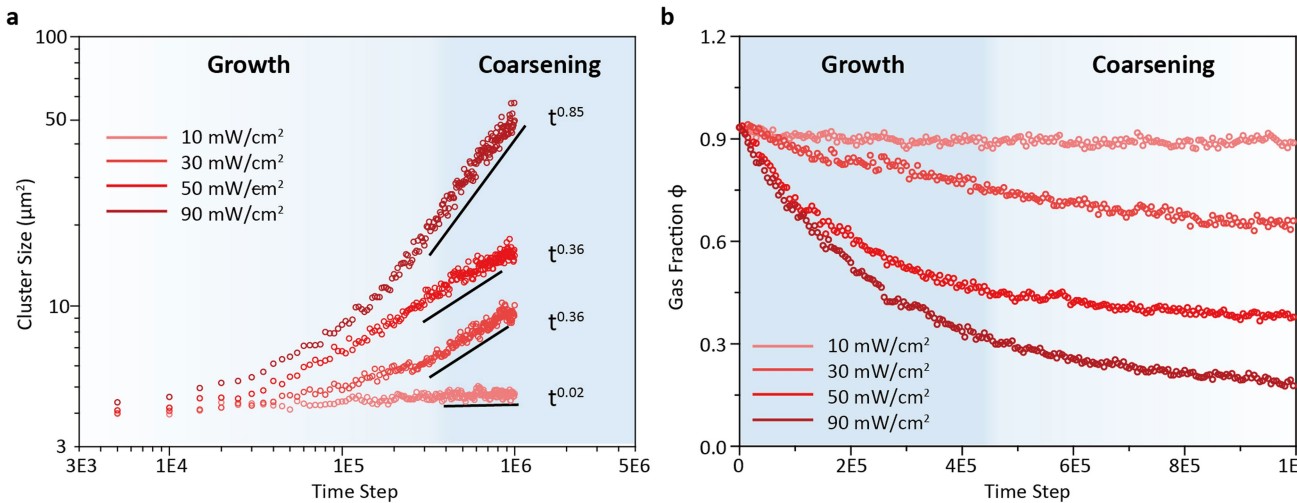

**Extended Data Fig. 8 | Brownian dynamics simulation of colloidal mixture.** **a**, The Brownian dynamics simulated cluster size evolution of the binary colloidal system under various apparent pair potentials fitted from experimental results under different red light illumination (640-660 nm, 10 mW/cm², 30 mW/cm², 50 mW/cm², 90 mW/cm²), corresponding to the Fig. 3b. The cluster sizes are compatible with the relation of $S(t) \sim t^{0.02}$, $S(t) \sim t^{0.36}$, $S(t) \sim t^{0.36}$, $S(t) \sim t^{0.85}$, respectively. **b**, The simulated gas fraction of binary mixture decreases over time under various apparent pair potential fitting from experimental results.