## [Peer Review File · Nature]

Manuscript Title: Photochromism from wavelength-selective colloidal phase segregation

Reviewer Comments & Author Rebuttals

Reviewer Reports on the Initial Version:

Referee #1 (Remarks to the Author):

This work describes how TiO₂ colloids loaded with different dyes respond to external light to (1) initiate a redox process in the surrounding solution; (2) generate wavelength-dependent attractive hydrodynamic interactions; and (3) mix and demix on demand, including formation of layers which, to an external user, appear as a color-changing display.

The work is certainly well done, both technically and theoretically – in fact, the models in the SI are well formulated and capture the key aspects of the experimental system with regards to the wavelength-dependent aggregation phenomena (although the explanation of the 3D layering effect appears much less rigorous and additional details would certainly be helpful).

From a fundamental point of view, this paper will definitely be of interest to the active colloids community.

This said, I am somewhat hesitant to suggest acceptance because I do not quite see the wider applicability of the phenomena described. In particular, while the authors envisage applications as displays or active camouflage systems, I note that the separation/mixing processes requires 30-100 mW/cm² of power at a given spectral range. For comparison, on a sunny day, the sunlight delivers ca. 100 mW/cm² but this is over the entire spectral range. In other words, the power of say, red, or blue light within Sun's spectrum will be much less and perhaps in single-digit mW/cm². This means that the idea of camouflage system is highly unlikely to work in practice. Furthermore, the times of irradiation to achieve color switching are quite long (e.g., 2 min in Figure 6) which makes such devices impractical as battlefield camouflage or, even worse, displays. These performance characteristics are much worse than thermochromic materials or Janus-colloid (e-ink systems) and fall somewhat short of a really transformative Nature paper. Thus, I will leave this as a question to the authors whether they could significantly improve these parameters? If so, I would be very happy to recommend acceptance.

Two minor comments relate to the structure of the paper (which should be shortened and more to the point from the very beginning) and also to the conceptual novelty (vis a vis ref 47 in which the same team reported TiO₂-based, dye-sensitized swimmers, though not in the context of phase separation).

Referee #2 (Remarks to the Author):

It is a very nice work about photoactive colloidal particles, where the authors demonstrated the

phase segregations of a colloidal system composed of three types of TiO₂ colloidal species coded with distinctive dyes due to the photoinduced particle-particle interactions. As these colloidal particles have different light reflectance and absorbing properties, realizing controllable segregation of different chromatic particles means that macroscopic colorful display may be tuned based on these particles, which is called as “photochromic nanoswarms”, and the related kinetic mechanism was also studied experimentally and theoretically. The whole manuscript is well organized and interesting for reading. The results are robust, interesting and attractive. I think this manuscript has the potential to draw broad interest from the audience of Nature. However, several major issues need to be addressed before the consideration of its publication:

1. I think the special point in this work is the on-demand segregation of the developed photoactive colloidal particles, as light-powered micro-swimmers have been already developed (which, at least those of TiO₂ motors should be cited herein). The power source was identified to the induced-redox reaction of the dye-sensitized TiO₂ particle, which created the diffusiophoresis and diffusioosmosis flow on the neighboring particles. However, the used light sources are narrow bandwidth without UV region, so how did the redox reaction happen? The dye is fully covered on the TiO₂, how does the photo-induced charge transfer?
2. Does the TiO₂ play an essential role involved in the photo-active process? Are there any other species that can also act as the core materials? How does the particle size affect the 3D phase separation? Why did the authors choose magenta, yellow and cyan-colored TiO₂ particles to demonstrate the concept? Can the authors realize RGB primary colors of active TiO₂ colloids?
3. Does the surface wettability of the particles affect the interaction between particles?
4. The authors confirm that the apparent pair potential between two active particles are stronger than the active-passive pair. However, according to Fig. 3a-b, Video S-1 and calculated results Video S-2, almost every aggregation of active particles contains passive particles, sitting at the central zone or interior zone. What is the kinetic reason for this phenomenon? It seems irreconcilable with the proposed mechanism.
5. From Fig.4c, the selection of the excitation light wavelength is somehow arbitrary. May the authors activate only one type of TiO₂ colloidal species via the selection of a proper wavelength (e.g., the wavelength at the highest absorbance of each dye)?
6. How did the authors determine the flux of H⁺ and OH⁻ from the redox reactions? What is the difference in the flux of H⁺ and OH⁻ from the illuminated and shadowed surface of TiO₂ particles with different colors. The propagation of the excitation light with different wavelengths in dye-sensitized TiO₂ particles would be helpful to support the above settings.
7. For highly concentrated nanoswarm mixture, I don't think the movement of particles is the same with that in dilute system. The friction between particles may take a significant role, which may enable the expected demixing not to happen. May the authors investigate the effects of particles concentration on the performances?
8. What will happen if the light is illuminated from the bottom and the side? Does the demixing phenomenon and phase segregation appear in 3D directions?
9. May the authors comment on the possibility of using invisible light (e.g., UV or IR) to build photochromic active colloids with stable reflective colors like fluorescent materials? It is better if the authors present some experimental results (UV activation is expectable).
10. In Fig. 5c and d, why are yellow particles are concentrated in the middle? Is it possible for the authors to improve the homogeneity of the layered colloid?
11. May the authors investigate the influence of the proportions of different colored TiO₂ particles

on the photochromic properties of the active colloids under various colored lights? Is it possible for the authors to discuss the cases if one or two species are activated.

12. What is about the heat production in the active colloids? How to exclude the influence from photo-induced thermal effect? This may affect the accuracy and resolution of motion, and thus the phase segregations of particles and color stability after long-term light illumination.

13. The color brightness in Fig. 6f is low. How to improve the color brightness?

14. The colored image will gradually diminish within 30 minutes in the dark due to the remixing of colloidal particles. The authors attribute this to the Brownian motion. I don't think it is a good explanation. Can those particles after separation be re-active again? Cycling experiments should be conducted to clarify the stability of the whole processing. The kinetics of the photochromic active colloids over the illumination time (in Fig. 5 or Fig. 6) should be investigated.

15. More details should be provided to understand the experimental color gamut of the photochromic nanoswarms. At least provide some demonstrations including displays with and without nanoswarm for comparison. In addition, the authors are suggested to present a concept-of-demonstration how to use this programmable photochromic ink to active optical camouflage, and give some descriptions about the advantages over the current photochromic materials.

Referee #3 (Remarks to the Author):

The authors have created a chromo-responsive colloidal system in which inter-particle interactions were tuned as a function of the wavelength of light and its intensity. They were able to selectively segregate types of particles by creating colloids with different dyes which could be chemically activated by an external light source. They calculated the size and stability of the segregated phases. Finally, they probed the adaptability of a system of three different dye-sensitized colloids as a proxy to camouflage behavior. The layered segregation can potentially have important applications.

The experiments are cleverly and carefully designed. Overall, the work is very creative. However, I find the theoretical analysis very poor, and the simulations not convincing. Although the snapshots in Fig. 3 a and b are similar, the experimental power laws of cluster size with time in Fig. 3c do not resemble those obtained with the simulations in Fig. S7a. Furthermore, there is no discussion on why different power laws arise in the cluster size versus time. Are those power laws true asymptotic behaviors? Moreover, there is no analysis of the distribution of clusters sizes, to determine the degree of polydispersity in cluster size and/or the possibility of self-similarity. A discussion of why Brownian dynamics with the fitting potential should describe the system in 2D would be valuable. The supplementary information is well-written. It is not clear, however, why Eqs. (8) and (9) are derived if they are not used. Please provide the relevance of these equations to the analysis.

Minor points

1. Line 202: I think the active particles are blue and passive are red.
2. There is no consistency in their terminology; it fluctuates between segregation and demixing, please choose one (lines 140 and 144).
3. The authors frequently remind the readers that SQ2, LEG4, and L0 are dyes that reflect cyan,

magenta, and yellow, respectively. On this note, it could be helpful to clarify that the colors used to classify the different dye-sensitized colloids in the figures are those that the dyes reflect and not absorb.

4. There are plenty of small grammatical mistakes which should be revised.

Author Rebuttals to Initial Comments:

Referees' comments (#1):

This work describes how TiO₂ colloids loaded with different dyes respond to external light to (1) initiate a redox process in the surrounding solution; (2) generate wavelength-dependent attractive hydrodynamic interactions; and (3) mix and demix on demand, including formation of layers which, to an external user, appear as a color-changing display.

The work is certainly well done, both technically and theoretically – in fact, the models in the SI are well formulated and capture the key aspects of the experimental system with regards to the wavelength-dependent aggregation phenomena (although the explanation of the 3D layering effect appears much less rigorous and additional details would certainly be helpful). From a fundamental point of view, this paper will definitely be of interest to the active colloids community.

This said, I am somewhat hesitant to suggest acceptance because I do not quite see the wider applicability of the phenomena described. In particular, while the authors envisage applications as displays or active camouflage systems, I note that the separation/mixing processes requires 30-100 mW/cm² of power at a given spectral range. For comparison, on a sunny day, the sunlight delivers ca. 100 mW/cm² but this is over the entire spectral range. In other words, the power of say, red, or blue light within Sun's spectrum will be much less and perhaps in single-digit mW/cm². This means that the idea of camouflage system is highly unlikely to work in practice. Furthermore, the times of irradiation to achieve color switching are quite long (e.g., 2 min in Figure 6) which makes such devices impractical as battlefield camouflage or, even worse, displays. These performance characteristics are much worse than thermochromic materials or Janus-colloid (e-ink systems) and fall somewhat short of a really transformative Nature paper. Thus, I will leave this as a question to the authors whether they could significantly improve these parameters? If so, I would be very happy to recommend acceptance.

Response to Reviewer #1:

We appreciate the referee's positive and valuable comment, and we agree that the improved photochromic performance, including faster response and higher sensitivity, would be essential for practical application. We are surely working on this aspect, as will be described in this revision.

On the other hand, in our original manuscript and experiment, the main target is to introduce a new method for optically controllable colloidal systems, which we believe is an ideal system for researchers in colloidal science and soft matter physics as research tools. As a result, we simply applied a commercially available light source with high intensity for photochromic demonstration purposes while neither qualitatively studying the photochromic performance of the colloidal solution nor putting much effort into improving the response time and sensitivity. In this new revision, we quantified photochromic performance with more careful study and tried to optimize the formula of the active ink by adjusting the particle size and concentration, which resulted in a much-improved photochromic performance and image quality. However, in the current stage, the system still requires $\sim 10\text{-}20\text{mW/cm}^2$ light intensity to function properly. We attribute this not to the fundamental limit of the method; instead, this limited sensitivity is due to the high gravitational force of TiO_2 particles preventing efficient 3D layering and the lower absorption coefficient of the commercial dyes. As such, further improvement is expected by using particles with lower density (for example, Polystyrene- TiO_2 core-shell particles) and designing new dyes with high absorption coefficients (like many organometallic dyes). Since switching to new material system may result disconnection from the data in the first part of the paper, we believe this potential improvement is out of the scope of this paper. We only focus on optimizing the existing dye- TiO_2 system in this paper, while further investigation will be reported in later study.

On the other hand, the performance of our active ink is in pair or better than state-of-the-art photochromic materials and could be further improved. I hope you may agree that the photochromic active nanoswarm has much broader potential beyond current photochromic materials. Please find more details point-by-point response as follows.

Q1. The explanation of the 3D layering effect appears much less rigorous and additional details would certainly be helpful.

We agree that the 3D layers process should be better understood, which is essential for further improvement of this photochromic ink. In this revised version, we applied more fine-grained and coarse-grained simulations to elucidate the details of this 3D phenomenon. First, the COMSOL Multiphysics is used to simulate the detailed flow field in matrix with active and passive particles, where all particles experienced higher light intensity on the top than the bottom due to scattering and shadowing of the particle matrix. As shown in Figure R1a (Figure S10a), upon illumination, a vertical flow is generated between particles by diffusiophoresis, which is the origin of the attractive potential. Under this attractive potential, active particles clustered together, and the vertical flows overlapped and intensified (Figure R1b (Figure S10b)).

In 3D phase segregation experiment, there are roughly 30-50 layers of particles in the colloidal solution, and the lower particles serve as the pseudo-substrate for the upper layers. As shown in Figure R1c (Figure S10c), when passive particles (blue) settle below a layer of active particles (red), the upwards electroosmotic flow is generated, similar to the particle monolayer on the glass substrate. As more layers of active particles stack, the upwards flow further, which brings passive particles to the top, while the active particles sediment to the bottom due to the counteraction of the upward fluid flow generation (Figure R1d, R1e (Figure S10d, S10e)).

Fig. R1 | **a-b**, Flow field simulation around active particles (red) and passive particles (blue) shows **a**, inward fluid flow that pushes the particles together when well separated and **b**, the upward fluid flow intensified after cluster formation. **c-e**, Simulations of fluid flow on particles under different conditions. **c**: Active particles located on multiple layers of passive particles, referring to the initial state of the active particles; **d**: One active particle suspended in solution, moving downwards due to the self-generated upward flow in light gradient; **e**: Active particles located below passive particles, referring to the final state of the active particles. **f**. Evolution of active (red) and passive (blue) particles in the binary mixture from 3D Brownian dynamics simulations.

We further elucidate this 3D layering process in crowd condition with three-dimensional Brownian dynamics simulations, where the light-dependent Morse potential, $u = D_0[e^{-2\alpha(r-r_0)} - 2e^{-\alpha(r-r_0)}]$ describing the particle interaction and the light-dependent vertical force describing the upwards flow field is adopted. Here, the parameters of the potential function, i.e., the depth of the potential well D_0 and α controlling the 'width' of the potential are obtained from the apparent pair potentials, as measured in our experiment (Figure 2). In addition, a height-dependent vertical force proportional to particle activity is also

applied to reflect the vertical flow field. As shown in Figure R1f (Figure S10f), initially, it is assumed that active and passive particles are uniformly distributed in the simulated domain. Under the mentioned field as generated by light illumination, active and passive particles spontaneously separated vertically, which is consistent with our experimental observation.

Q2. In particular, while the authors envisage applications as displays or active camouflage systems, I note that the separation/mixing processes requires 30-100 mW/cm² of power at a given spectral range. For comparison, on a sunny day, the sunlight delivers ca. 100 mW/cm² but this is over the entire spectral range. In other words, the power of say, red, or blue light within Sun's spectrum will be much less and perhaps in single-digit mW/cm². This means that the idea of camouflage system is highly unlikely to work in practice. Furthermore, the times of irradiation to achieve color switching are quite long (e.g., 2 min in Figure 6) which makes such devices impractical as battlefield camouflage or, even worse, displays. These performance characteristics are much worse than thermochromic materials or Janus-colloid (e-ink systems) and fall somewhat short of a really transformative Nature paper.

Firstly, we appreciate this valuable comment, and we agree that the improved performance, including faster response and higher sensitivity, would be essential for practical application. In our previous version, we did not quantify and optimize the photochromic performances of swarm ink, and the 1 μm TiO₂ particles were utilized for demonstration purposes, which resulted in poor photochromic sensitivity and response time.

In this new revision, we quantified photochromic performance, and tried to optimize the sensitivity by adjusting the particle size and concentration, which yielded an improved response time, sensitivity, and image contrast.

In particular, nanoswarms composed of particles of three different sizes: 500 nm, 1 μm and 2 μm are investigated.

Fig. R2 | The performance of photochromic nanoswarm with various particle diameters. **a**, The optical images of the emerged university logo of different particle sizes. Scale bar: 2mm. **b**, Characteristic curve of different photochromic nanoswarm under different light intensity and time.

As shown in Figure R2 (Figure S12), compared to the previous version, where nanoswarm ink with 1 μm TiO₂ is used (middle Figure R2a), 500nm particles resulted in much-enhanced photosensitivity, where the brightness and contrast of the resulted image also significantly enhanced (left Figure R2a), while the larger particles show much deteriorated photochromic property (right Figure R2a). In addition, the previously inaccessible white color can also be obtained. To quantify the photosensitivity, characteristic curves of nanoswarm with 500 nm, 1 μm and 2 μm TiO₂ are measured and plotted as contrast versus light intensity and contrast versus exposure time. From the characteristic curves, we conclude that the threshold intensity and exposure time are around 10-20 mW/cm² and 1 min, respectively.

We also optimized the photochromic properties by adjusting the particle concentration. We investigated the photochromic property of colloidal ink with different particle concentrations. As shown in Figure R3 (Figure S13), the segregation proceeds in low-

concentration ink, while the image contrast is low as insufficient particles in the swarm. The nanoswarm image quality gradually improves with particle concentration, while an optimal concentration is achieved at $1 \times 10^{12}/\text{cm}^3$. Further increased particle concentration leads to deteriorated contrast and brightness, suggesting incomplete segregation due to significant friction and jamming between particles.

Fig. R3 | performance of photonic nanoswarm with different concentrations of the particle mixture. Scale bar: 2 mm.

We speculate that the photosensitivity is determined by the balance of upward flow generated by active particles and the gravity of passive particles, as shown in the simulation. As a result, the photosensitivity is not fundamentally limited by the working mechanism. Instead, lower density particles (such as high porosity TiO_2 ¹ or polystyrene- TiO_2 core-shell particles²) with less gravitational drag may further enhance nanoswarm photosensitivity. In addition, new photosensitive dyes with high absorption coefficient or better dye loading strategies, such as that demonstrated in recent publication³, which significantly enhance the loading efficiency with additives, may also enhance the photochromic performance of nanoswarm. However, while we are certainly working on this improvement, we believe those modifications deviate from this paper's focus too much and are not included in this manuscript.

On the other hand, the photochromic nanoswarm does offer some unique advantages as chromic materials. For example, since the colour image results from phase segregation, the response can be easily tuned by controlling the surface charge or doped with different dyes. Here, we applied positively charged TiO_2 to replace our original negatively charged ones. As shown in Figure R4 and Table R1 (Figure S14 and Table S4), a negative color image is achieved with positively charged particles instead of positive color images for negatively charged particles.

Considering the issues mentioned earlier, we have weakened the practical application in our manuscript, and the major highlight is the independent and continuous tunability of active states of each component in our multi-component systems

Fig. R4| Positive and negative imaging of photochromic nanoswarm modulated by zeta potential. Scale bar: 2 mm.

Table R1. The zeta potential of pristine TiO₂ and different dye loaded TiO₂

Zeta Potential (mV)				
	pristine	L0-TiO ₂	LEG4-TiO ₂	SQ2-TiO ₂
(+)TiO ₂	+10.33	+8.23	+7.78	+8.72
(-)TiO ₂	-38.45	-42.38	-48.68	-15.01

In addition, we compared the key features of current available photochromic materials, which are mainly based on the valence transition in material or photoisomerization of chromophore molecules. As shown in Table R3, the performance of our nanoswarm ink offers full-color rendering ability and immediate repatterning ability with high sensitivity and response speed.

Table R3. . Comparison of current state-of-the-art photochromic materials

Type of Materials	Excitation Light	Response Time	Display Color	Mechanism	Ref.
Photochromic Nanoswarm	Full Spectrum (~10-20 mW/cm ²)	1 min	Full Spectrum (positive and negative response) No Erase needed	Phase Segregation	This Paper
Phosphomolybdic Acid	365nm UV Flashlight (5 W)	5 min	Dark Blue Air oxidation erase	Valence Transition	[9]
Poly(ionic liquids) and Polyoxometalates	Projector Lamp (210 W)	2 min	Light Yellow and Bluish Green Air oxidation erase	Valence Transition	[10]
Bismuth Oxyhalide	365nm UV (6 mW/cm ²)	1 min	Brown Black Air oxidation erase	Valence Transition	[11]
WO ₃ Nanoparticles	UV lamp (5 W)	3 min	Blue Heat erase	Valence Transition	[12]
Photochromic Diarylethene Derivatives	254nm UV (1 mW/cm ²)	30 min	Cyan or Magenta or Yellow Visible light erase	Isomerization	[13]
Photochromic Dyes	Full Spectrum (1200 Lumens)	20-120 min	Full Spectrum (positive response) UV (4W) erase	Isomerization	[14]

Q3. Two minor comments relate to the structure of the paper (which should be shortened and more to the point from the very beginning) and also to the conceptual novelty (vis a vis ref 47 in which the same team reported TiO₂-based, dye-sensitized swimmers, though not in the context of phase separation).

Thank you for your suggestion; we have shortened the paper.

Reviewer #2 (Remarks to the Author):

It is a very nice work about photoactive colloidal particles, where the authors demonstrated the phase segregations of a colloidal system composed of three types of TiO₂ colloidal species coded with distinctive dyes due to the photoinduced particle-particle interactions. As these colloidal particles have different light reflectance and absorbing properties, realizing controllable segregation of different chromatic particles means that macroscopic colorful display may be tuned based on these particles, which is called as "photochromic nanoswarms", and the related kinetic mechanism was also studied experimentally and theoretically. The whole manuscript is well organized and interesting for reading. The results are robust, interesting and attractive. I think this manuscript has the potential to draw broad interest from the audience of Nature.

However, several major issues need to be addressed before the consideration of its publication:

Response to Reviewer #2:

We appreciated the positive comment from the referee, and thanks for the valuable suggestions. We have strengthened the manuscript with more theoretical explanations as well as newly improved the photochromic nanoswarm. Please check the point-by-point revision lists as follows.

Q1. I think the special point in this work is the on-demand segregation of the developed photoactive colloidal particles, as light-powered micro-swimmers have been already developed (which, at least those of TiO₂ motors should be cited herein). The power source was identified to the induced-redox reaction of the dye-sensitized TiO₂ particle, which created the diffusiophoresis and diffusi-osmosis flow on the neighboring particles. However, the used light sources are narrow bandwidth without UV region, so how did the redox reaction happen? The dye is fully covered on the TiO₂, how does the photoinduced charge transfer?

We agree that more research articles about TiO₂ motors should be cited, some related references have been added to the manuscript (Ref. 41-42).

The absorption spectrum of TiO₂ is mainly in UV region (bandgap 3.0 eV), but after being sensitized with organic dyes, the absorption spectrum is determined by the dye absorption. The principle of sensitized TiO₂ active colloidal is similar to that of the well-studied dye-sensitized solar cell (DSSC) (Figure R5). The visible photon first excites the electrons in the HOMO of the dye molecules onto the LUMO and subsequently transfers into the conduction band (CB) of TiO₂ as the LUMO of the designed dye molecules is higher than TiO₂'s CB. As a result, the absorption spectrum of TiO₂ is no longer limited by the bandgap of TiO₂, which enables the colloidal to be sensitive to visible and NIR radiation.

Fig. R5 | Schematic of electron transport in the dye-sensitized solar cell.

Q2. Does the TiO₂ play an essential role involved in the photo-active process? Are there any other species that can also act as the core materials? How does the particle size affect the 3D phase separation? Why did the authors choose magenta, yellow and cyan-colored TiO₂

particles to demonstrate the concept? Can the authors realize RGB primary colors of active TiO₂ colloids?

Thanks for these comments and great suggestions, which enables improved photochromic properties of our nanoswarm. The photo-active process is related to the DSSC mechanism mentioned above. For this reason, many semiconductor materials such as TiO₂, ZnO, SnO₂ that has been used in DSSC is a potential candidate to act as the loading materials in our photochromic nanoswarm. On the other hand, since we used commercial solar dyes optimized for the conduction band of TiO₂, the efficiency is better for TiO₂ colloids. We did test the activity of dye-loaded ZnO, SnO₂, and BaTiO₃ particles, which all show visible light activity similar to TiO₂. However, since the activity is not as good as TiO₂, we did not include them in our paper. In principle, other semiconductor materials can be used, while different dyes should be designed. To leverage the rich knowledge in DSSC, TiO₂ is still the material of choice, which is low-cost, biocompatible, and highly stable.

As for the effect of particle size, we greatly thank the referee for this suggestion, which enabled a much-improved performance of the nanoswarm ink. We investigated the performance of photochromic nanoswarm with three different particle sizes: 500 nm, 1 μm and 2 μm.

Fig. R2 | The performance of photochromic nanoswarm with various particle diameters. **a**, The optical images of the emerged university logo of different particle sizes. Scale bar: 2mm. **b**, Characteristic curve of different photochromic nanoswarm under different light intensity and time.

As shown in Figure R2 (Figure S12), compared to our demonstration in the previous version (middle Figure R2a), smaller particles (500 nm) resulted in much-enhanced photosensitivity, where the brightness and contrast of the resulted image improved, while the larger particles (2 μm) shows much deteriorated photochromic property. In addition, the previously inaccessible white colour can also be obtained. To quantify the photosensitivity, characteristic curves of nanoswarm with 500 nm, 1 μm and 2 μm TiO_2 are measured and plotted as contrast versus light intensity and exposure time. From the characteristic curves, we conclude that the minimum light intensity and exposure time are 20 mW/cm^2 and 1 min, respectively.

We speculate that the photosensitivity is determined by the balance of upward flow generated by the active particle and the gravity of the passive particle. As a result, the photosensitivity is not fundamentally limited by the working mechanism. Instead, lower density particles (such as high porosity TiO_2 ¹ or polystyrene- TiO_2 core-shell particles²) with less gravitational drag and higher activity particles with better dye loading may further enhance nanoswarm photosensitivity.

The reason to choose CMY combination instead of RGB is that RGB is used in additive colour, such as OLED or LCD displays, where the combination of RGB light yields white. In contrast, our system relies on reflective light, which requires CMY as primary for subtractive colour as commonly used in printing and ink; the combination of CMY is black (Figure R6).

Fig. R6 | RGB Model and CMYK Model.

Q3. Does the surface wettability of the particles affect the interaction between particles?

Thank you for your comment. In principle, the surface wettability will change the particle interaction in the short range as the depletion force^{4,5}. However, the interaction probed in this research is much longer ranged over the micrometer scale; the molecular level interaction, such as the wettability effect, is negligible. In addition, our particle needs to be hydrophilic enough to maintain the dispersion in water, as the hydrophobic particles suffer from colloidal percolation due to high surface energy. On the other hand, surface modification will also change the reactivity. For example, surface coating agents like polymers or functionalized silane species will influence the diffusion process, especially when the polymer is closely packed on the surface due to hydrophobicity, blocking the reaction.

We have tested PVP and Pluronic F127 as hydrophilic agents and trichloro(1H,1H,2H,2H-perfluorooctyl)silane as the hydrophobic modifier. Polymer modification greatly diminishes the segregation ability, and silane modification reduces the photoactivity of the particle. One feasible route is to synthesize hydrophilic/hydrophobic polyelectrolytes that can support the charge transfer process, while it is beyond the scope of this manuscript.

Q4. The authors confirm that the apparent pair potential between two active particles are stronger than the active-passive pair. However, according to Fig. 3a-b, Video S-1 and calculated results Video S-2, almost every aggregation of active particles contains passive particles, sitting at the central zone or interior zone. What is the kinetic reason for this phenomenon? It seems irreconcilable with the proposed mechanism.

We appreciate the careful check from the referee, and noticed this incomplete segregation. In this revision, we developed a theoretical thermodynamic phase diagram of our mixture, which is consistent with the experimental observation. We attributed the observed incomplete segregation to this thermodynamics and the kinetics of the colloidal solution.

From thermodynamic point of view, the binary mixture phase diagram can be generated by considering the $\Delta_{mix}G$, which is fundamentally determined by the dimensionless

parameter $\xi = \frac{z}{k_B T} (u_{AB} - \frac{u_{AA} + u_{BB}}{2})$, where u_{AA} , u_{BB} , and u_{AB} represent the interaction potential of AA, BB, and AB, respectively, and the z is the effective coordination number. Similar to the traditional binary mixture shown in Figure R7a, when $\xi > 2$, the overall effective free energy has a double minimum, which defines the stable phase, while the $\frac{\partial^2 \Delta_{mix} G}{\partial x^2} = 0$ defines the metastable phases. In our system, the ξ can be easily adjusted by tuning the incident light intensity (Figure R7b), as opposed to the traditional system where the ξ can only be adjusted by temperature.

As shown in the phase diagram (Figure R7b), the mole fractions on the stable line and metastable line are not 1, which means the segregation is not 100% as prevented by the entropy penalty. Experimentally, our observation also agrees with this prediction that the phase-segregated clusters are in metastable phases (points in Figure R7b).

In addition, as the particle size increases, the thermal fluctuation is slowed down, which may lead to thermodynamically unstable phases as passive particles are trapped inside the matrix of active particles. This is particularly true in high light intensity cases, where the particle interaction is strong, which suggests very slow relaxation to stable regions.

To summarize, incomplete phase segregation is a result of thermodynamic minimization under lower light intensity and kinetic trapping in higher light intensity, which aligns with the experimental observation.

Fig. R7 | a, The ξ variation of the Gibbs energy of the binary mixture. When $\xi > 2$, the binary mixture forms two phases with compositions corresponding to the two local minima of the curve. **b**, The theoretical phase diagram of the active binary mixture with respect to red and blue light intensities (left axis) and ξ (right axis). Inset points: the obtained phase

compositions in experiment, where red and blue points represent the phase compositions with red and blue light illumination, respectively.

Q5. From Fig.4c, the selection of the excitation light wavelength is somehow arbitrary. May the authors activate only one type of TiO₂ colloidal species via the selection of a proper wavelength (e.g., the wavelength at the highest absorbance of each dye)?

Thank you for this comment, and we agree that it is desirable to have distinct excitation to particle species. However, to satisfy the CMY colour system, the selection of dye absorption spectrum is limited. By searching the commercially available dyes, our dye selection is close to optimal, while there is still a significant overlap for L0 and LEG4 at 400-500 nm as shown in Figure R8 (Figure 4c).

Fig. R8 | The normalized absorbance of ethanolic solution of dyes (L0 (yellow), LEG4 (magenta), and SQ2 (cyan)) with distinctive absorption spectra covering the visible spectrum. Inset: the spectral range of illuminating RGB light source.

With this dye selection, in principle, it is possible to selectively active three particles with 400-420 nm (L0), 480—500 nm (LEG4) and 640-660 nm (SQ2). To prevent the photodegradation of dye by TiO₂, UV light must be avoided. In our current setup, where the LED light source with FWHM around 50 nm is used, we leave a sufficient safe margin in our wavelength selection to avoid UV exposure. In addition, to realize the macroscopic photochromic nanoswarm, a commercial projector is selected, which requires the match of our wavelength selection to the emission of the commercial projector (Xe Lamp) to provide enough illumination to the colloidal ink.

To summarize, selective activation is possible for our photochromic colloidal system by selecting different light sources or designing other dyes, while in this paper, a compromise is made to match the properties of readily available dyes and commercial projector light source.

Q6. How did the authors determine the flux of H^+ and OH^- from the redox reactions? What is the difference in the flux of H^+ and OH^- from the illuminated and shadowed surface of TiO_2 particles with different colors. The propagation of the excitation light with different wavelengths in dye-sensitized TiO_2 particles would be helpful to support the above settings.

Thank you for your comment; we have modified our model after careful investigation. As shown in Figure R9, the light propagation and absorption on an individual particle are evaluated by finite-difference time-domain (FDTD) simulation, while the absolute absorption coefficient is measured experimentally by 3D reconstruction microscopy. In our case, the light penetration depth of the particle is around 500 nm, which suggests a unity absorption of light in the particle as seen in the particle. In our previous experiment, we evaluated the quantum efficiency of the dyed particle system with photocurrent solar cell measurement⁶, which allowed up to calculate the H^+ and OH^- flux on the particle.

Similar to our previous report⁷, the illuminated hemisphere act as a photoanode, where H^+ is generated, $QH_2 \rightarrow BQ + 2H^+$, and the shadowed surface act as a photocathode, where OH^- is generated, $BQ + 2H_2O \rightarrow QH_2 + 2OH^-$, which created electric field around the particle. Such self-generated electric field will drive fluid flow that drags the particles by ionic diffusiophoresis.

Fig. R9 | FDTD simulation of light power absorption of a dye loaded particle with excitation light illuminated from the top.

Q7. For highly concentrated nanoswarm mixture, I don't think the movement of particles is the same with that in dilute system. The friction between particles may take a significant role, which may enable the expected demixing not to happen. May the authors investigate the effects of particles concentration on the performances?

Thank you for the comment, and we agree that particle concentration should play an important role in the segregation process. Due to thermal fluctuation, the nanoswarm behaves as a liquid, where interstitial space allows the particle rearrangement for segregation. On the other hand, this interstitial space shrinks as particle concentration increases, which prevents the demixing process. We investigated the photochromic property of colloidal ink with different particle concentrations. As shown in Figure R3 (Figure S13), the segregation proceeds in low-concentration ink, while the image contrast is low as there are insufficient particles in the swarm. The nanoswarm image quality gradually improves with particle concentration, while an optimal concentration is achieved at $1 \times 10^{12} / \text{cm}^3$. Further increased particle concentration leads to deteriorated contrast and brightness, suggesting incomplete segregation due to significant friction between particles.

Fig. R3 | performance of photonic nanoswarm with different concentrations of the particle mixture. Scale bar: 2 mm.

Q8. What will happen if the light is illuminated from the bottom and the side? Does the demixing phenomenon and phase segregation appear in 3D directions?

Thanks for the suggestion. When the light is illuminated from the bottom, the observed phase segregation is similar to illumination from the top, where the coloured image appears at the bottom ink surface, as shown in figure R10a.

When the light is illuminated from the side, the in-plane particle segregation is observed. To show this in-plane phase segregation, we illuminate blue light from one side of the chamber and observe the phase segregation under the microscope. As shown in figure R10b, the horizontal phase separation is observed, which supports the proposed 3D layering process. Therefore, the demixing phenomenon and phase segregation can appear in any direction.

On the other hand, since the concentrated nanoswarm is not transparent, we cannot observe the 3D segregation process directly. We speculated that the 3D segregation should be similar to our observation in 2D and in diluted ink in 3D, while the light scattering and particle rearrangement could be more complicated and subject to future investigation.

Fig. R10 | **a**, The university logo appears at the bottom when patterned by bottom illumination. Scale bar: 2 mm. **b**, Horizontal phase segregation with blue light illumination from the side. Scale bar: 5 μ m.

Q9. May the authors comment on the possibility of using invisible light (e.g., UV or IR) to build photochromic active colloids with stable reflective colors like fluorescent materials? It is better if the authors present some experimental results (UV activation is expectable).

The dyed TiO₂ particles can indeed respond to invisible light, including UV and IR. However, as stated previously, the organic dye molecules can be photobleached under UV irradiation with photolysis on TiO₂. On the other hand, IR response is possible, as many readily available infrared dyes have been developed. To demonstrate this, we select MK245⁸, which is sensitive

to near-infrared light. After mixing with LEG4-TiO₂ colloids, the MK245-TiO₂ colloids aggregate under IR light (750nm) irradiation, making the red LEG4-TiO₂ colloids flow upward, and the red patterns appear in the swarm (Figure R11).

Fig. R11 | Pattern emergence under NIR illumination in LEG4-TiO₂ and MK245-TiO₂ mixture. Scale bar: 2 mm.

Q10. In Fig. 5c and d, why are yellow particles are concentrated in the middle? Is it possible for the authors to improve the homogeneity of the layered colloid?

We appreciate that the referee has carefully examined our data and agree that the homogeneity of the layered colloids should be improved. We redid the experiment with improved uniformity of the colloids, and new confocal images have been obtained (Figure R12) to replace the previous Figure 5b-d.

Fig. R12 | New confocal images with higher homogeneity. Scale bar: 50 μ m.

Q11. May the authors investigate the influence of the proportions of different colored TiO₂ particles on the photochromic properties of the active colloids under various colored lights? Is it possible for the authors to discuss the cases if one or two species are activated.

We agree that various proportions of dye-sensitized TiO₂ colloids will result in various photochromic responses. Generally, when the fraction of an ingredient decreases, the colour response to this ingredient diminishes. In our experiment, we tried to balance the colour of different pigment particles to realize the desired colour response. For example, when the ratio of L0-sensitized TiO₂ colloids is reduced to a quarter of the normal amount, the yellow and

green response is much deteriorated compared to the HKU logo-based demo (Figure R13). One flexibility of the proposed chromic ink system is that the colour response can be easily tuned with different formulations.

Fig. R13 | performance of the photonic nanoswarm by reducing the amount of L0-TiO₂ ingredient. Scale bar: 2 mm.

For nanoswarm composed of three colloidal species, there are six possible excitation recombinations (except all excitation and no excitation). As shown in Figure R14 (Figure 5d), we showed the effect of selectively activating one or two species in ink. To better understand it, a table is provided where six illumination conditions related to six activation combinations (Table R2), resulting in six colour blocks.

Fig. R14 | The emerged colour blocks on photochromic ink after 2 min exposure. Inset: The exposure pattern with six colour blocks containing red, green, blue, cyan, magenta, and yellow. Scale bar: 2 mm.

Table R2. Expected layered phase behaviour under different illumination spectra.

Illuminated light	Light component	Top layer (passive)	Bottom layer (active)
Red	Red	L0-TiO ₂ + LEG4-TiO ₂	SQ2-TiO ₂
Green	Green	L0-TiO ₂ + SQ2-TiO ₂	LEG4-TiO ₂
Blue	Blue	SQ2-TiO ₂	L0-TiO ₂ + LEG4-TiO ₂
Cyan	Green + Blue	SQ2-TiO ₂	L0-TiO ₂ + LEG4-TiO ₂
Magenta	Red + Blue	LEG4-TiO ₂ (weak active)	L0-TiO ₂ + SQ2-TiO ₂
Yellow	Red + Green	L0-TiO ₂	LEG4-TiO ₂ + SQ2-TiO ₂

Q12. What is about the heat production in the active colloids? How to exclude the influence from photoinduced thermal effect? This may affect the accuracy and resolution of motion, and thus the phase segregations of particles and color stability after long-term light illumination.

In our system, the interaction between colloids is resulted from photoinduced diffusiophoresis and diffusioosmosis, while the thermal effect is negligible. Much higher light intensity will be required for significant temperature change and photothermal effect. To demonstrate this, we remove the redox couple in the solution and perform both the microscopic illumination and macroscopic pattern experiment. In the absence of redox couples, neither phase segregation nor macroscopic image formation is observed.

Q13. The color brightness in Fig. 6f is low. How to improve the color brightness?

Thank you for this comment. To improve the image brightness, we optimized the formula with smaller particles and optimized concentration. As shown in Figure R15, this new formula greatly improves the image brightness and sensitivity of nanoswarm.

We believe the nanoswarm photosensitivity is determined by the balance of upward flow from active particles and the gravity of passive particles. As a result, the photosensitivity is not fundamentally limited by the working mechanism. We expect lower density particles (for example, more porous particles or PS-TiO₂ core-shell particles) or higher activity particles (for example, with different dyes and better loading strategy) can further enhance the photochromic performance. However, while we are certainly working on this improvement, this modification deviates from the main focus of this paper and is not included in this paper.

Fig. R15 | Comparison of the university logo obtained previously (**a**, with $1\mu\text{m TiO}_2$) and in this revision (**b**, with 500 nm TiO_2). Scale bar: 2mm.

Q14. The colored image will gradually diminish within 30 minutes in the dark due to the remixing of colloidal particles. The authors attribute this to the Brownian motion. I don't think it is a good explanation. Can those particles after separation be re-active again? Cycling experiments should be conducted to clarify the stability of the whole processing. The kinetics of the photochromic active colloids over the illumination time (in Figure 5 or Figure 6) should be investigated.

We agree that reversible patterning ability is an essential feature for practical application. As shown in Figure R16 (Figure 5f), several different paintings are selected to show the reversible patterning ability of nanoswarm. In this experiment, the exposure time for each picture is 2 mins. The colour image lasts around 30-60 mins without further exposure patterning. In contrast, the new images can be directly patterned on existing patterns without noticeable sensitivity change. This proves that the image fading is not due to the chemical degradation of the nanoswarm ink. Thus, the only possible reason for natural fading would be the remixing of colloidal particles with Brownian diffusion.

To study the kinetics of the photochromic ink over illumination time, we quantify the photosensitivity. As shown in Figure R2b (Figure S12b), characteristic curves of nanoswarm with 500 nm , $1\mu\text{m}$ and $2\mu\text{m TiO}_2$ are measured and plotted as contrast versus light intensity and exposure time.

Fig. R16 | Sequential patterning of the nanoswarm with different color paintings with 2 min exposure. Inset: the original projected patterns. Scale bar: 2 mm.

Q15. More details should be provided to understand the experimental color gamut of the photochromic nanoswarms. At least provide some demonstrations including displays with and

without nanoswarm for comparison. In addition, the authors are suggested to present a concept-of-demonstration how to use this programmable photochromic ink to active optical camouflage, and give some descriptions about the advantages over the current photochromic materials.

Thank you for this comment. We have provided detailed information on the experimental colour gamut measurements in the Supplementary Information. Firstly, a series of images can be obtained by projecting six colour blocks (Figure R14) with different times and intensities. Then all the samples are placed under the pinhole of the integrating sphere (Figure R17), and the simulated sunlight is illuminated into the integrating sphere. The spectrometer and commercial software (Oceanview) are employed to draw the received reflected light spectrum into the colour gamut diagram to obtain the colour gamut.

Fig. R17 | Schematic of color gamut measurement.

Existing photochromic materials can change their color when exposed to ultraviolet or visible light due to the photochemical generation of new chromophore or valence transition⁹⁻¹⁴. As a result, only one color can be achieved with given molecules. In contrast, as discussed above, the photochromic nanoswarm is highly programmable by tuning the composition, ratio, and zeta potential of the dye-loaded TiO₂ colloids. Compared with photochromic molecules, our system does not rely on generating new chromophores, which intrinsically have much better fatigue resistance, which is a major concern in photochromic molecules. Also, our mechanism enables better flexibility as new photoresponse can be easily achieved without designing new molecular structures. The sensitivity of our current system is in pair or better than existing photochromic materials (Table R3, Table S5).

On the other hand, although improved, the sensitivity of the current nanoswarm is still lower than desired. As stated previously, we are still working on further enhancing the sensitivity of the nanoswarm with new particles and new dyes. The legit demonstration of active optical camouflage will be reported with higher sensitivity formula as proposed in the paper. We have changed the description in the paper accordingly.

Table R3. . Comparison of current state-of-the-art photochromic materials .

Type of Materials	Excitation Light	Response Time	Display Color	Mechanism	Ref.
Photochromic Nanoswarm	Full Spectrum (~10-20 mW/cm ²)	1 min	Full Spectrum (positive and negative response) No Erase needed	Phase Segregation	This Paper
Phosphomolybdic Acid	365nm UV Flashlight (5 W)	5 min	Dark Blue Air oxidation erase	Valence Transition	[9]
Poly(ionic liquids) and Polyoxometalates	Projector Lamp (210 W)	2 min	Light Yellow and Bluish Green Air oxidation erase	Valence Transition	[10]
Bismuth Oxyhalide	365nm UV (6 mW/cm ²)	1 min	Brown Black Air oxidation erase	Valence Transition	[11]
WO ₃ Nanoparticles	UV lamp (5 W)	3 min	Blue Heat erase	Valence Transition	[12]
Photochromic Diarylethene Derivatives	254nm UV (1 mW/cm ²)	30 min	Cyan or Magenta or Yellow Visible light erase	Isomerization	[13]
Photochromic Dyes	Full Spectrum (1200 Lumens)	20-120 min	Full Spectrum (positive response) UV (4W) erase	Isomerization	[14]

Reviewer #3(Remarks to the Author):

The authors have created a chromo-responsive colloidal system in which inter-particle interactions were tuned as a function of the wavelength of light and its intensity. They were able to selectively segregate types of particles by creating colloids with different dyes which could be chemically activated by an external light source. They calculated the size and stability of the segregated phases. Finally, they probed the adaptability of a system of three different dye-sensitized colloids as a proxy to camouflage behavior. The layered segregation can potentially have important applications.

The experiments are cleverly and carefully designed. Overall, the work is very creative.

Response to Reviewer #3:

We thank the reviewer for the positive comments and revised the manuscript based on the suggestions and comments. Please check our point-to-point response as follows.

Q1. The experiments are cleverly and carefully designed. Overall, the work is very creative. However, I find the theoretical analysis very poor, and the simulations not convincing. Although the snapshots in Fig. 3 a and b are similar, the experimental power laws of cluster size with time in Fig. 3c do not resemble those obtained with the simulations in Fig. S7a. Furthermore, there is no discussion on why different power laws arise in the cluster size versus time. Are those power laws true asymptotic behaviors? Moreover, there is no analysis of the distribution of clusters sizes, to determine the degree of polydispersity in cluster size and/or the possibility of self-similarity.

We thank the reviewer for the positive comments and careful check. The power law growth of the cluster size is ubiquitously observed in the phase separation process and has been well described in the literature¹⁵⁻¹⁷. Here, we assume the cause of the power law growth is at least similar to those observed in traditional equilibrium phase separation systems. In a thermodynamic system, the power law-dependent exponent should be related to the growth mechanism. We revisited our previous power law calculation and found that the previous asymptotic fitting was not correct. As shown in revised Figure R18 (Figure 3c and Figure S7a), we extracted the coarsening stage exponents ν of colloids segregation under different illumination. Under mid-range light intensity (30 mW/cm², 50 mW/cm²), the growth exponents are stable $\sim 1/3$ and increased to ~ 0.5 under strong illumination (90 mW/cm²), suggesting additional growth mechanism may play roles as the system biased far away from equilibrium. A similar exponent increase has been reported previously in active and passive system^{17,18}, while further investigation will be needed to elucidate the exact cause in our light activated mixture.

We believe this discrepancy under strong bias might be caused by uncertainties in simulations, including the damping factor, and the Morse potential. In our simulation, we adopted the Morse potential which is commonly used in Molecular Dynamic studies. However, although the standard Morse potential fits well with the particle-particle interaction potential in long-range, it is merely an empirical estimation of experiments and may deviate significantly in the short-range. As a result, under high light intensity, the particle-particle distance is small, where the exponent's deviation is expected.

Fig. R18 | **a**, Experimental result of cluster size evolution of the binary colloidal. **b**, The molecular dynamics simulated cluster size evolution of the binary colloidal system under various apparent pair potentials fitted from experimental results, corresponding to Figure R18a.

To verify the topological self-similarity during the phase segregation, the chord length distribution function^{17,19} $P(l/\langle l \rangle)$ is measured for different temporal snapshot. Briefly, a straight line is randomly generated across the image. The chord length l is then determined by the length of the line segment inside the cluster. By varying the starting point and the orientation of the straight line randomly, a series of l from different straight lines are obtained and then normalized by its mean value (characteristic length) $\langle l \rangle$, from which yield the chord length distribution function $P(l/\langle l \rangle)$.

As shown in Figure R19 (Figure S8), $P(l/\langle l \rangle)$ is independent of time, indicating the self-similarity during the cluster growth, which is also the origin of the power law dependence and very similar to the phase separation in classical thermodynamic mixture.

Fig. R19 | The chord length distribution functions for the colloid-poor phase at different times (300 s, 360 s, 420 s and 480 s) after being scaled by the characteristic length $\langle l \rangle$.

In addition, we have strengthened theoretical explanations in the manuscript to help deepen understanding and further improvement.

First, in this revised version, we applied more fine-grained and coarse-grained simulations to elucidate the details of this 3D layering segregation. The COMSOL Multiphysics is used to simulate the detailed flow field in particle matrix, where all particles receive higher light intensity on the top than the bottom, due to scattering and shadowing of the particle matrix. As shown in Figure R1a (Figure S10a), upon illumination, a vertical flow is generated between particles by diffusiophoresis, which is the origin of the attractive potential. Under this attractive potential, active particles clustered together, and the vertical flows overlapped with each other and intensified (Figure R1b (Figure S10b)).

In 3D phase separation experiment, there are roughly 30-50 layers of particles in the colloidal solution, and the lower particles serve as the pseudo-substrate for the upper layers. As shown in Figure R1c (Figure S10c), when passive particles (blue) settle below a layer of active particles (red), the upwards electroosmotic flow is generated, similar to the particle monolayer on the glass substrate. As more layers of active particles stack, the upwards flow further, which brings passive particles to the top, while the active particles sediment to the bottom due to the counteraction of the upward fluid flow generation (Figure R1d, R1e (Figure S10d, S10e)).

Fig. R1| a-b, Flow field simulation around active particles (red) and passive (blue) particles shows **a**, inward fluid flow that pushes the particles together when well separated and **b**, the upward fluid flow intensified after cluster formation. **c-e**, Simulations of fluid flow on particles under different conditions. **c**: Active particles located on multiple layers of passive particles, referring to the initial state of the active particles; **d**: One active particle suspended in solution, moving downwards due to the self-generated upward flow in light gradient; **e**: Active particles located below passive particles, referring to the final state of the active particles. **f**. Evolution of active (red) and passive (blue) particles in the binary mixture from 3D Brownian dynamics simulations.

We further elucidate this 3D layering process in crowd condition with three-dimensional Brownian dynamics simulations, where the light-dependent Morse potential describing the particle interaction and the light-dependent vertical force describing the upwards flow field is adopted. Initially, it is assumed that active and passive particles are uniformly distributed in the simulated domain. As shown in Figure R1f (Figure S10f), active and passive particles spontaneously separated under light illumination, consistent with our experimental observation.

Second, we strengthen the analysis of the thermodynamics of the colloidal solution with theoretical phase diagram. From the thermodynamic point of view, the binary mixture phase diagram can be generated by considering the $\Delta_{mix}G$, which is fundamentally determined by the dimensionless parameter $\xi = \frac{z}{k_B T} \left(u_{AB} - \frac{u_{AA} + u_{BB}}{2} \right)$, where u_{AA} , u_{BB} , and u_{AB} represent the interaction potential of AA, BB, and AB, respectively, the z is the effective coordination number. Similar to the traditional simple binary mixture shown in Figure R7a, when $\xi > 2$, the overall effective free energy has a double minimum, which defines the stable phase, while the $\frac{\partial^2 \Delta_{mix}G}{\partial x^2} = 0$ defines the metastable phases. In our system, the ξ can be easily adjusted by tuning the incident light intensity (Figure R7b), as opposed to the traditional system where the ξ can only be modified by temperature.

As shown in the phase diagram (Figure R7b), our observation agrees well with this prediction and suggests that the active colloidal mixture is in metastable phases (points in Figure R7b).

Fig. R7 | a, The ξ variation of the Gibbs energy of the binary mixture. When $\xi > 2$, the binary mixture forms two phases with compositions corresponding to the two local minima of the curve. **b**, The theoretical phase diagram of the active binary mixture with respect to red and blue light intensities (left axis) and ξ (right axis). Inset points: the obtained phase compositions in experiment, where red and blue points represent the phase compositions with red and blue light illumination, respectively.

Q2. A discussion of why Brownian dynamics with the fitting potential should describe the system in 2D would be valuable. The supplementary information is well-written. It is not clear,

however, why Eqs. (8) and (9) are derived if they are not used. Please provide the relevance of these equations to the analysis.

Thank you for the comment. We have deleted Eq. (8) and Eq. (9) and rewrote the second part in supplementary information: simulation of electric field and fluid flow.

Q3. Minor points

1. Line 202: I think the active particles are blue and passive are red.

Thanks for the careful checking. This is indeed a point easily confuses the reader. In the experiment part, under red light illumination, the active particles are blue in appearance (as red light is absorbed), and passive ones are red, while in the simulation part, following a habit, we set the active particles red and passive particles blue to show which light the particle absorbs. We have marked this in the captions.

2. There is no consistency in their terminology; it fluctuates between segregation and demixing, please choose one (lines 140 and 144).

We appreciate this comment and have used segregation uniformly throughout the whole manuscript.

3. The authors frequently remind the readers that SQ2, LEG4, and L0 are dyes that reflect cyan, magenta, and yellow, respectively. On this note, it could be helpful to clarify that the colors used to classify the different dye-sensitized colloids in the figures are those that the dyes reflect and not absorb.

Thank you for the comment. To make it clear, we have made legends in the figure.

4. There are plenty of small grammatical mistakes which should be revised.

Thanks for this comment. We have further revised our manuscript carefully.

Reference

- 1 Liu, Y. *et al.* Radially oriented mesoporous TiO₂ microspheres with single-crystal-like anatase walls for high-efficiency optoelectronic devices. *Sci. Adv.* **1**, e1500166 (2015).
- 2 Xue, Y., Wang, F., Luo, H. & Zhu, J. Preparation of Noniridescent Structurally Colored PS@TiO₂ and Air@C@TiO₂ Core-Shell Nanoparticles with Enhanced Color Stability. *ACS Appl. Mater. Interfaces* **11**, 34355-34363 (2019).
- 3 Ren, Y. *et al.* Hydroxamic acid preadsorption raises efficiency of cosensitized solar cells. *Nature* (2022).
- 4 Young, K. L. *et al.* Assembly of reconfigurable one-dimensional colloidal superlattices due to a synergy of fundamental nanoscale forces. *Proc. Natl. Acad. Sci. U. S. A.* **109**, 2240-2245 (2012).
- 5 Kraft, D. J. *et al.* Surface roughness directed self-assembly of patchy particles into colloidal micelles. *Proc. Natl. Acad. Sci. U. S. A.* **109**, 10787-10792 (2012).
- 6 Dai, J. *et al.* Solution-Synthesized Multifunctional Janus Nanotree Microswimmer. *Adv. Funct. Mater.* **31**, 2106204 (2021).
- 7 Zheng, J. *et al.* Orthogonal navigation of multiple visible-light-driven artificial microswimmers. *Nat. Commun.* **8**, 1438 (2017).
- 8 Pydzińska, K. & Ziółek, M. Solar cells sensitized with near-infrared absorbing dye: Problems with sunlight conversion efficiency revealed in ultrafast laser spectroscopy studies. *Dyes Pigm.* **122**, 272-279 (2015).
- 9 Fan, J. *et al.* Flexible, switchable and wearable image storage device based on light responsive textiles. *Chem. Eng. J.* **404**, 126488 (2021).
- 10 Wales, D. J. *et al.* 3D-Printable Photochromic Molecular Materials for Reversible Information Storage. *Adv. Mater.* **30**, 1800159 (2018).
- 11 Zhang, X. *et al.* Self-accelerating photocharge separation in BiOBr ultrathin nanosheets for boosting photoreversible color switching. *Chem. Eng. J.* **428**, 131235 (2022).
- 12 Dong, J. & Zhang, J. Photochromic and super anti-wetting coatings based on natural nanoclays. *J. Mater. Chem. A* **7**, 3120-3127 (2019).
- 13 Jeong, W. *et al.* Full Color Light Responsive Diarylethene Inks for Reusable Paper. *Adv. Funct. Mater.* **26**, 5230-5238 (2016).
- 14 Jin, Y. *et al.* Photo-Chromeleon: Re-Programmable Multi-Color Textures Using Photochromic Dyes. *ACM*, 701-712 (2019).
- 15 Laradji, M., Toxvaerd, S. & Mouritsen, O. G. Molecular Dynamics Simulation of Spinodal Decomposition in Three-Dimensional Binary Fluids. *Phys. Rev. Lett.* **77**, 2253-2256 (1996).
- 16 Bastea, S. & Lebowitz, J. L. Spinodal Decomposition in Binary Gases. *Phys. Rev. Lett.* **78**, 3499-3502 (1997).
- 17 Tateno, M. & Tanaka, H. Power-law coarsening in network-forming phase separation governed by mechanical relaxation. *Nat. Commun.* **12**, 912 (2021).
- 18 Redner, G. S., Hagan, M. F. & Baskaran, A. Structure and Dynamics of a Phase-Separating Active Colloidal Fluid. *Phys. Rev. Lett.* **110**, 055701 (2013).
- 19 Lu, B. & Torquato, S. Chord-length and free-path distribution functions for many-body systems. *J. Chem. Phys.* **98**, 6472-6482 (1993).

Reviewer Reports on the First Revision:

Referee #1 (Remarks to the Author):

In my previous report, I praised the overall "coolness" of the paper but also pointed out several aspects that would need improvement, especially regarding the performance characteristics. I understand certain parameters cannot be really tuned (e.g., time response) -- in this aspect, I accept the authors' rationale that competing photochromic systems work on similar timescales. This said, I am very happy the authors managed to significantly improve other parameters (e.g., contrast) by optimizing their system with respect to particle size and particle concentration. Glancing over the responses to the other two reviewers, I see that the theoretical basis has also been improved. At this point I have no significant scientific objections. I still wonder a bit whether this paper might better fit, say, Nature Materials, and whether the novelty is pristine (vis a vis similar works from the same authors) but these may be just my singular views -- in essence, I have no objections for the work to be published as revised.

Referee #2 (Remarks to the Author):

Tang and co-authors reported two impressive merits of light-driven TiO₂ micro/nanomotor systems, even considering the previous publications of ref. 49 and Research 2022, 2022, 9816562 (which is suggested to be cited herein). One is the phase segregation phenomenon induced by external light with various wavelength of the mixture of the above different coded microswimmers. The other is the potential applications as displays or active camouflage systems. In this revision, the authors have perfectly addressed most of the concerns from me and others, except a few listed in the below. The work certainly represents a giant breakthrough in the field of micro/nanomotors and active colloids. I think it deserves the publication by Nature if the authors further confirm the impressive application performances.

1. The response time of the system is still in the scale of minutes. Is it possible to lower the response time to be around milliseconds making it more suitable for commercial displays? The author may outlook some potential measures or provide a theoretical calculation or simulation.
2. It is important for reflective displays and active camouflage systems to realize bistable or multi-stable states of patterns if energy-saving and surrounding conditions (e.g. sun-light, temperature) are considered. What will happen for the patterns in Fig. 5e and f, if the light is switched off? Will they gradually become faded or obscured due to the Brownian motion? May the authors further provide some data about the time evolution of the patterns if the light is switched off.
3. Can the authors provide the CIE color space diagram they could achieve as well as the greyscale for this system?
4. What is the standard reference for 100% intensity during the reflection spectra measurements? Were the reflectance spectra normalized against a mirror or a white Lambertian diffuser? This information is suggested to include in the characterizations section. The absolute reflectivity rather than normalized ones of the system is suggested to provide for the understanding of the intensity that this strategy could achieve.

Referee #3 (Remarks to the Author):

The authors had made a great effort to respond to the comments of the reviewers. The paper demonstrates a very interesting phenomenon with possibility of applications. I found the introduction and motivation of the work scattered. The work is relevant to active matter and has applications in colored inks (maybe with improving time responses in colored displays) and not because colloids can be considered giant atoms or because it can be used to explain phase segregation in various systems. For example, the introduction can easily move from just the first sentences with 1 to 4 references to the second paragraph references 15-20 instead of some irrelevant work to this study as follows:

The macroscopic properties of materials are fundamentally determined by interactions between their basic composition units. In mixtures of molecules with similar interactions, the mixing entropy dominates and leads to a well-mixed solution, whereas a distinct intermolecular interaction leads to the enthalpy penalty and causes phase segregation¹⁻⁴. Classical paths to phase segregation in colloidal mixtures have been demonstrated by changing thermodynamic variables such as temperature and/or solvent interactions¹⁵⁻²⁰. Active colloids offer a new approach for realizing complex phase behaviors²⁵⁻²⁸, which are deeply rooted in the nonequilibrium nature of the systems. Proposed models include Motility-Induced Phase Separation²⁹⁻³³, where the dispersed self-propulsion particles condense due to particles' mobility and repulsive interaction. Theoretically, the active colloidal mixture can self-phase separate due to distinctive diffusivity³³, temperature³⁴, and activity^{35,36}.

There are clearly many more things to explore or explain better in this work that maybe for future studies. The COMSOL simulations are well-done and justified. However, the discussion on power laws in Fig. 3 is still weak. For example, in the previous version there was a power law regime of growth nearly linear in time (something that has been explained by hydrodynamic flow when the interface between domains is sharp if both segregated phases are fluid) but now it is closer to 0.5. Could it be just which could be due to an extended transient regime between the classical $1/3$ growth and a hydrodynamic grow? In any case, the phenomenon is well-documented and I therefore support its publication.

Author Rebuttals to First Revision:

Referee #1:

In my previous report, I praised the overall "coolness" of the paper but also pointed out several aspects that would need improvement, especially regarding the performance characteristics. I understand certain parameters cannot be really tuned (e.g., time response) -- in this aspect, I accept the authors' rationale that competing photochromic systems work on similar timescales. This said, I am very happy the authors managed to significantly improve other parameters (e.g., contrast) by optimizing their system with respect to particle size and particle concentration. Glancing over the responses to the other two reviewers, I see that the theoretical basis has also been improved. At this point I have no significant scientific objections. I still wonder a bit whether this paper might better fit, say, Nature Materials, and whether the novelty is pristine (vis a vis similar works from the same authors) but these may be just my singular views -- in essence, I have no objections for the work to be published as revised.

We thank the reviewer for positive comments.

Referee #2:

Tang and co-authors reported two impressive merits of light-driven TiO₂ micro/nanomotor systems, even considering the previous publications of ref. 49 and Research 2022, 2022, 9816562 (which is suggested to be cited herein). One is the phase segregation phenomenon induced by external light with various wavelength of the mixture of the above different coded microswimmers. The other is the potential applications as displays or active camouflage systems. In this revision, the authors have perfectly addressed most of the concerns from me and others, except a few listed in the below. The work certainly represents a giant breakthrough in the field of micro/nanomotors and active colloids. I think it deserves the publication by Nature if the authors further confirm the impressive application performances.

We thank the reviewer for positive comments and revised the manuscript based on the suggestions and comments. Please check our point-to-point response as follows.

1. The response time of the system is still in the scale of minutes. Is it possible to lower the response time to be around milliseconds making it more suitable for commercial displays? The author may outlook some potential measures or provide a theoretical calculation or simulation.

We agree that the improved photochromic performance, including faster response and higher sensitivity, would be essential for wide application. As stated in the revision, we optimize the photochromic performance by adjusting the particle size and concentration, while further improvement is possible by significantly change the particle system, such as using lower density Polystyrene-TiO₂ core-shell particles and applying new dyes with higher absorption coefficients (like many organometallic dyes). However, this change defers too far from the current system, and not suitable to put in a single publication. On the other hand, a simple estimation can be used to consider the possible response time. As the typical swimming speed of photoactive particle is $\sim 10\mu\text{m/s}$, assuming 10-50 μm solution thickness will be needed to reach desired optical density, the response time close to 1 second is possible, which is paired with the response time of the current e-ink system ($\sim 1000\text{ms}$), while millisecond maybe too challenging. In this sense, this technology should not be expected as a replacement for LCD or OLED display. Instead, as a completely passive material, it may target applications with low response time requirements, such as e-reader or optical camouflage.

The related discussion is added in the main text of the manuscript.

2. It is important for reflective displays and active camouflage systems to realize bistable or multi-stable states of patterns if energy-saving and surrounding conditions (e.g. sun-light, temperature) are considered. What will happen for the patterns in Fig. 5e and f, if the light is switched off? Will they gradually become faded or obscured due to the Brownian motion? May the authors further provide some data about the time evolution of the patterns if the light is switched off.

The patterns will gradually diminish within 40 min if the light is switched off (as shown in Figure R1), due to the remixing of colloidal particles with the Brownian motion, which suggested reconfigurability of this photochromic swarm. As for the biostability, the similar technique used in e-ink system can be applied to our system, while the detailed formula and realization is subject to further development.

Fig. R1 | The sequential images show the stability of the light painting HKU logo for 0 min, 10 min, 20 min, 30 min and 40 min, respectively. Scale bar: 2 mm.

3. Can the authors provide the CIE color space diagram they could achieve as well as the greyscale for this system?

The CIE color space diagram has been included in the Figure S6d. In display, each elementary RGB color has eight bits ($2^8 = 256$ tones, from 0 to 255) of data. To study the colorscale of the colloidal swarm ink, three arrays of red, green and blue light patterns with different gradient from 0 to 255 with interval at 16 were projected sequentially as shown in Figure R2a. After 2 min exposure, three arrays of display patterns appeared, showing gradual increase in brightness corresponding to the incident stimulus pattern (Figure R2b).

Fig. S6d | Experimental color gamut of the photochromic nanoswarm presented in a standard CIE-1931 color space.

Fig. R2 | **a**, Optical patterns with red, green and blue light of different gradient from 0 to 255 with interval at 16. **b**, The macroscopic image of acquired color patterns under the illumination of array light in (a). Scale bar: 2 mm.

4. What is the standard reference for 100% intensity during the reflection spectra measurements? Were the reflectance spectra normalized against a mirror or a white Lambertian diffuser? This information is suggested to include in the characterizations section. The absolute reflectivity rather than normalized ones of the system is suggested to provide for the understanding of the intensity that this strategy could achieve.

In the reflection spectra measurements, the reflectance spectra normalized against a white Lambertian diffuser (PTFE), which has been added to the method section.

Referee #3 :

The authors had made a great effort to respond to the comments of the reviewers. The paper demonstrates a very interesting phenomenon with possibility of applications. I found the introduction and motivation of the work scattered. The work is relevant to active matter and has applications in colored inks (maybe with improving time responses in colored displays) and not because colloids can be considered giant atoms or because it can be used to explain phase segregation in various systems. For example, the introduction can easily move from just the first sentences with 1 to 4 references to the second paragraph references 15-20 instead of some irrelevant work to this study as follows:

The macroscopic properties of materials are fundamentally determined by interactions between their basic composition units. In mixtures of molecules with similar interactions, the mixing entropy dominates and leads to a well-mixed solution, whereas a distinct intermolecular interaction leads to the enthalpy penalty and causes phase segregation¹⁻⁴. Classical paths to phase segregation in colloidal mixtures have been demonstrated by changing thermodynamic variables such as temperature and/or solvent interactions¹⁵⁻²⁰. Active colloids offer a new approach for realizing complex phase behaviors²⁵⁻²⁸, which are deeply rooted in the nonequilibrium nature of the systems. Proposed models include Motility-Induced Phase Separation²⁹⁻³³, where the dispersed self-propulsion particles condense due to particles' mobility and repulsive interaction. Theoretically, the active colloidal mixture can self-phase separate due to distinctive diffusivity³³, temperature³⁴, and activity^{35,36}.

There are clearly many more things to explore or explain better in this work that maybe for future studies. The COMSOL simulations are well-done and justified. However, the discussion on power laws in Fig. 3 is still weak. For example, in the previous version there was a power law regime of growth nearly linear in time (something that has been explained by hydrodynamic flow when the interface between domains is sharp if both segregated phases are fluid) but now it is closer to 0.5. Could it be just which could be due to an extended transient regime between the classical $1/3$ growth and a hydrodynamic grow? In any case, the phenomenon is well-documented and I therefore support its publication.

We appreciate the reviewer for positive comments and specific revision suggestions, and we accept that the introduction and motivation should be more focused, so we revised the manuscript. For the deviation of the power law, some additional growth mechanism may play roles as the system biased far away from equilibrium, or as the reviewer suspect, there is an extended transient regime between the classical $1/3$ growth and a hydrodynamic grow, but we do not have definitive evidence yet.